# Exercise and aerobic capacity in individuals with spinal cord injury: A systematic review with meta-analysis and meta-regression

Daniel D. Hodgkiss[1], Gurjeet S. Bhangu[2,3], Carole Lunny[4], Catherine R. Jutzeler[5,6], Shin-Yi Chiou[1,7,8,9], Matthias Walter[2,10], Samuel J. E. Lucas[1,7], Andrei V. Krassioukov[2,11,12], Tom E. Nightingale[1,2,9]*

1 School of Sport, Exercise and Rehabilitation Sciences, University of Birmingham, Birmingham, United Kingdom, 2 International Collaboration on Repair Discoveries (ICORD), University of British Columbia, Vancouver, Canada, 3 MD Undergraduate Program, Faculty of Medicine, University of British Columbia, Vancouver, Canada, 4 Knowledge Translation Program, Li Ka Shing Knowledge Institute, St. Michael's Hospital, Toronto, and University of British Columbia, Vancouver, Canada, 5 Department of Health Sciences and Technology, ETH Zurich, Zurich, Switzerland, 6 Schulthess Clinic, Zurich, Switzerland, 7 Centre for Human Brain Health, University of Birmingham, Birmingham, United Kingdom, 8 MRC Versus Arthritis Centre for Musculoskeletal Ageing Research, University of Birmingham, Birmingham, United Kingdom, 9 Centre for Trauma Science Research, University of Birmingham, Birmingham, United Kingdom, 10 Department of Urology, University Hospital Basel, University of Basel, Basel, Switzerland, 11 Department of Medicine, Division of Physical Medicine and Rehabilitation, University of British Columbia, Vancouver, Canada, 12 GF Strong Rehabilitation Centre, Vancouver Coastal Health, Vancouver, Canada

* T.E.Nightingale@bham.ac.uk

**Data Availability Statement:** All code/scripts and data used for the analysis are available at https://github.com/jutzca/Exercise-and-fitness-in-SCI

## Abstract

### Background

A low level of cardiorespiratory fitness [CRF; defined as peak oxygen uptake ($\dot{V}O_{2peak}$) or peak power output (PPO)] is a widely reported consequence of spinal cord injury (SCI) and a major risk factor associated with chronic disease. However, CRF can be modified by exercise. This systematic review with meta-analysis and meta-regression aimed to assess whether certain SCI characteristics and/or specific exercise considerations are moderators of changes in CRF.

### Methods and findings

Databases (MEDLINE, EMBASE, CENTRAL, and Web of Science) were searched from inception to March 2023. A primary meta-analysis was conducted including randomised controlled trials (RCTs; exercise interventions lasting >2 weeks relative to control groups). A secondary meta-analysis pooled independent exercise interventions >2 weeks from longitudinal pre-post and RCT studies to explore whether subgroup differences in injury characteristics and/or exercise intervention parameters explained CRF changes. Further analyses included cohort, cross-sectional, and observational study designs. Outcome measures of interest were absolute ($A\dot{V}O_{2peak}$) or relative $\dot{V}O_{2peak}$ ($R\dot{V}O_{2peak}$), and/or PPO. Bias/quality was assessed via The Cochrane Risk of Bias 2 and the National Institute of Health Quality Assessment Tools. Certainty of the evidence was assessed using the Grading of

**Funding:** This work was supported by a Nathalie Rose Barr PhD studentship awarded by the International Spinal Research Trust (#NRB123 to DDH), a Flexible and Enhanced Learning Program at the Faculty of Medicine within the University of British Columbia (to GSB), the Canadian Institutes of Health Research (Grant to CL), the Swiss National Science Foundation (Ambizione Grant PZ00P3_186101 to CRJ), the Wings for Life Research Foundation (2020_118 to CRJ), and Michael Smith Foundation for Health Research Trainee Awards in conjunction with the International Collaboration on Repair Discoveries and Rick Hansen Foundation, respectively (17110 to MW; 17767 to TEN). The funders had no role in study design, data collection and analysis, decision to publish, or preparation of the manuscript.

**Competing interests:** The authors have declared that no competing interests exist.

**Abbreviations:** 1RM, one repetition maximum; ACE, arm-crank ergometry; ACSM, American College of Sports Medicine; AIS, American Spinal Injury Association Impairment Scale; $A\dot{V}O_{2peak}$, absolute peak oxygen uptake; CENTRAL, Cochrane Central Register of Controlled Trials; CERT, Consensus on Exercise Reporting Template; CI, confidence interval; CPET, cardiopulmonary exercise test; CRF, cardiorespiratory fitness; CVD, cardiovascular disease; EMBASE, Excerpta Medica Database; FES, functional electrical stimulation; GRADE, Grading of Recommendations, Assessment, Development and Evaluation; HIIT, high-intensity interval training; HR, heart rate; HRmax, maximum heart rate (age-predicted); HRpeak, peak heart rate; HRR, heart rate reserve; IQR, interquartile range; ISNCSCI, International Standards for Neurological Classification of Spinal Cord Injury; LTPA, leisure time physical activity; MET, metabolic equivalent; MTP, maximal tolerated power; MVPA, moderate-to-vigorous physical activity; NHLBI, National Heart, Lung and Blood Institute; NIH, National Institutes of Health; PPO, peak power output; PRISMA, Preferred Reporting Items for Systematic Reviews; RCT, randomised-controlled trial; RER, respiratory exchange ratio; RoB 2, the Cochrane Risk of Bias 2 tool; RPE, rating of perceived exertion; $R\dot{V}O_{2peak}$, relative peak oxygen uptake; SCI, spinal cord injury; SCS, spinal cord stimulation; SD, standard deviation; TIDieR, Template for Intervention Description and Replication; TSI, time since injury; $\dot{V}O_2$, oxygen uptake; $\dot{V}O_{2peak}$, peak oxygen uptake; $\dot{V}O_{2reserve}$, reserve oxygen uptake; W, watts; WMD, weighted mean difference.

Recommendations Assessment, Development and Evaluation (GRADE) approach. Random effects models were used in all meta-analyses and meta-regressions.

Of 21,020 identified records, 120 studies comprising 29 RCTs, 67 pre-post studies, 11 cohort, 7 cross-sectional, and 6 observational studies were included. The primary meta-analysis revealed significant improvements in $A\dot{V}O_{2peak}$ [0.16 (0.07, 0.25) L/min], $R\dot{V}O_{2peak}$ [2.9 (1.8, 3.9) mL/kg/min], and PPO [9 (5, 14) W] with exercise, relative to controls ($p < 0.001$). Ninety-six studies (117 independent exercise interventions comprising 1,331 adults with SCI) were included in the secondary, pooled meta-analysis which demonstrated significant increases in $A\dot{V}O_{2peak}$ [0.22 (0.17, 0.26) L/min], $R\dot{V}O_{2peak}$ [2.8 (2.2, 3.3) mL/kg/min], and PPO [11 (9, 13) W] ($p < 0.001$) following exercise interventions. There were subgroup differences for $R\dot{V}O_{2peak}$ based on exercise modality ($p = 0.002$) and intervention length ($p = 0.01$), but there were no differences for $A\dot{V}O_{2peak}$. There were subgroup differences ($p \leq 0.018$) for PPO based on time since injury, neurological level of injury, exercise modality, and frequency. The meta-regression found that studies with a higher mean age of participants were associated with smaller changes in $A\dot{V}O_{2peak}$ and $R\dot{V}O_{2peak}$ ($p < 0.10$). GRADE indicated a moderate level of certainty in the estimated effect for $R\dot{V}O_{2peak}$, but low levels for $A\dot{V}O_{2peak}$ and PPO. This review may be limited by the small number of RCTs, which prevented a subgroup analysis within this specific study design.

## Conclusions

Our primary meta-analysis confirms that performing exercise >2 weeks results in significant improvements to $A\dot{V}O_{2peak}$, $R\dot{V}O_{2peak}$, and PPO in individuals with SCI. The pooled meta-analysis subgroup comparisons identified that exercise interventions lasting up to 12 weeks yield the greatest change in $R\dot{V}O_{2peak}$. Upper-body aerobic exercise and resistance training also appear the most effective at improving $R\dot{V}O_{2peak}$ and PPO. Furthermore, acutely injured, individuals with paraplegia, exercising for ≥3 sessions/week will likely experience the greatest change in PPO. Ageing seemingly diminishes the adaptive CRF responses to exercise training in individuals with SCI.

## Registration

PROSPERO: CRD42018104342

---

### Author summary

#### Why was this research done?

- Individuals with spinal cord injury (SCI) typically exhibit low levels of cardiorespiratory fitness (CRF). As such, these individuals are at a higher risk for the development of chronic diseases in comparison to the non-injured population.

- The current SCI-specific exercise guidelines encourage moderate-to-vigorous intensity aerobic exercise 40 min per week for fitness benefits or 90 min per week for

cardiometabolic health benefits. Yet, others have suggested individuals with SCI should be achieving 150 min per week in line with non-injured population guidelines.

- This systematic review with meta-analysis and meta-regression aimed to identify whether specific injury characteristics (e.g., time, neurological level, or severity of injury) or exercise intervention parameters (e.g., modality, intensity, volume) result in the greatest changes in CRF in individuals with SCI.

## What did the researchers do and find?

- We searched for studies that investigated the effects of exercise interventions lasting longer than 2 weeks on changes in absolute and relative peak oxygen consumption and/or peak power output (PPO) in individuals with SCI. In total, we included 120 studies of various study designs: 29 randomised controlled trials (RCTs), 67 pre-post studies, 11 cohort comparisons, 7 cross-sectional studies, and 6 observational studies.

- The greatest changes in PPO may be achieved by individuals with acute SCI or para-plegia. Upper-body aerobic and resistance exercise were identified as the most optimal exercise modalities. Furthermore, there is a trend suggesting that prescribing moder-ate-to-vigorous intensity aerobic exercise using either a percentage of the individual's peak heart rate (HR) or oxygen consumption, for 3 or more sessions per week, will result in the greatest improvements in PPO.

- Our findings support the minimum 40 min of weekly moderate-to-vigorous intensity exercise recommended by the SCI-specific exercise guidelines to significantly improve fitness. However, while not statistically significant, a 2-fold greater improvement in PPO was shown for interventions with exercise performed ≥90 min/week in compari-son to ≥40 min/week. Cross-sectional comparisons also revealed that individuals with SCI performing higher levels of physical activity were associated with a higher CRF.

## What do these findings mean?

- Exercise interventions >2 weeks can significantly improve CRF in individuals with an SCI by a clinically meaningful change greater than one SCI adjusted metabolic equiva-lent (i.e., ≥2.7 mL/kg/min). A one metabolic equivalent improvement has been associ-ated with a reduction in cardiovascular-related mortality risk in non-injured individuals.

- Our findings indicate that certain participant/injury characteristics and exercise inter-vention parameters are moderators of the changes observed in CRF across studies. These factors should be considered in the design of future exercise interventions. Future research should consider: following SCI-specific reporting guidelines (ensuring transparency of reporting), investigating the dose-response relationship between exer-cise and CRF or the influence of exercise intensity in this population, and consider how different injury characteristics impact the benefits of exercise on CRF.

- The main limitation of this review was the lack of RCTs comparing changes in CRF following an exercise intervention relative to a control group. This prevented subgroup

comparisons in this study design specifically and therefore we pooled pre-post and RCT exercise interventions to explore these effects.

## 1. Introduction

Spinal cord injury (SCI) is a complex neurological condition, caused by trauma, disease, or degeneration, which results in sensory-motor deficits (i.e., paralysis or paresis) below the level of lesion and autonomic dysfunctions. Progressive physical deconditioning following injury results in increased health care utilisation, reliance on personal assistance services, and a greater predisposition towards developing chronic diseases [1,2]. Individuals with SCI are at an increased risk of stroke, cardiovascular disease (CVD), and type 2 diabetes mellitus compared to non-injured counterparts [3–5]. The elevated incidence of these conditions in people with SCI emphasises the need for targeted interventions to address modifiable risk factors for these chronic diseases, such as cardiorespiratory fitness (CRF). In clinical populations CRF is typically defined as an individual's peak oxygen uptake ($\dot{V}O_{2peak}$) or peak power output (PPO). $\dot{V}O_{2peak}$ and PPO are determined during graded cardiopulmonary exercise testing (CPET) to the point of volitional exhaustion and represents the integrated functioning of different bodily systems (pulmonary, cardiovascular, and skeletal) to uptake, transport, and utilise oxygen for metabolic processes [6]. A number of prospective studies have indicated that CRF is at least as important, if not more so, than other traditional CVD risk factors (e.g., obesity, hypertension, and smoking) and is strongly associated with mortality [7–12].

Low levels of CRF have been widely reported in the SCI population [13], with the between-person variability partially explained by the neurological level and severity of injury (i.e., lower CRF reported in individuals with tetraplegia) [14]. A large proportion of the variance in CRF is also explained by physical activity [15], which is reduced in the SCI population [16,17]. Performing regular physical activity and/or structured exercise has long been promoted for improving CRF in individuals with SCI [18,19]. In 2011, the first evidence-based exercise guidelines, specifically for individuals with SCI were developed [20], which stated that "for important fitness benefits, adults with SCI should engage in at least 20 minutes of moderate-to-vigorous-intensity aerobic activity and strength-training exercises 2 times per week". This guideline has since been updated, yet remains the same with regards to CRF benefits [21]. Although this implies adults with SCI can accrue fitness benefits from volumes of activity well below that promoted in the general population, others have advocated that adults with a physical disability [22,23] and individuals with SCI [24] should aim to perform at least 150 min of aerobic exercise per week. For additional health benefits, it has been suggested that adults should perform closer to 300 min per week of moderate-intensity physical activity [25,26]. While the current SCI-specific guidelines likely represent the "minimum" threshold required to achieve CRF benefits, it has been suggested that this creates an impression that individuals with SCI do not need to be as physically active as the general population [27]. The dose-response relationship between exercise volume and CRF improvements in individuals with SCI remains to be elucidated.

It is noteworthy that the aforementioned SCI-specific exercise guidelines utilise the terminology of "moderate-to-vigorous" to describe the desired exercise intensity. This is in contrast to accepted guidelines in the general population whereby moderate and vigorous-intensity exercise are distinguished from one another with specific thresholds (e.g., $\geq$150 min of

moderate-intensity or ≥75 min of vigorous-intensity activity per week) [23]. Exercise intervention intensity has been shown to influence the magnitude of change in CRF in patients undergoing cardiac rehabilitation [28,29]. The feasibility/effectiveness of higher intensity exercise is also currently a topical area of research in the SCI population [30–32]. There is the potential for vigorous-intensity exercise to be more time efficient or lead to superior health benefits, although the impact of different high-intensity interval training protocols on CRF in individuals with SCI compared to moderate-intensity exercise is yet to be determined. A recent systematic review identified that exercise interventions of a specific modality yield distinct changes in certain cardiometabolic health outcomes, but not other outcomes, in individuals with SCI [33]. This provides rationale for wanting to investigate the efficacy of different exercise modalities on CRF in this population. Consequently, a number of research questions requiring further attention include:

1. Do injury-specific characteristics (e.g., tetraplegia versus paraplegia, acute versus chronic injuries, motor-complete versus incomplete) mediate CRF responses to exercise?

2. What is the best intensity, frequency, and volume of weekly exercise?

3. Is there an optimal conditioning modality [e.g., upper-body aerobic exercise, resistance training, functional electrical stimulation (FES), hybrid or multimodal exercise interventions]?

To address these questions, we performed a systematic review with meta-analysis and meta-regression to investigate the impact of different exercise interventions on changes in CRF in individuals with SCI. Moreover, we gathered evidence to determine whether key moderators (e.g., participant/injury characteristics, intervention/study characteristics, and risk of bias) influence these intervention effects.

## 2. Methods

This review is reported as per the Preferred Reporting Items for Systematic Reviews and Meta-Analyses (PRISMA) guidelines [34] (S1 Checklist) and was prospectively registered (PROSPERO ID CRD42018104342). The primary meta-analysis of this review included randomised controlled trials (RCTs; exercise intervention versus a comparison control group) given their robust, high-quality, and precise study design that can reliably estimate causation. A secondary, pooled meta-analysis was also conducted that combined the intervention arms of RCTs with non-randomised study designs (RCTs and pre-post exercise interventions without a comparison control group). Further secondary meta-analyses were conducted that included cohort comparisons (e.g., physically inactive versus habitual exercisers) and observational studies (e.g., standard of care rehabilitation), along with RCTs that specifically compared the impact of different exercise intensities (e.g., low or moderate-intensity versus vigorous or supramaximal) on CRF outcomes. Lastly, we qualitatively reviewed cross-sectional studies that reported associations between physical activity and CRF outcomes.

### 2.1. Eligibility criteria

Studies were required to meet the following inclusion criteria: (1) Adult (≥18 years) participants; (2) any acquired (traumatic, infection, cancer) SCI (note, studies were included if >80% of the sample met these 2 aforementioned inclusion criteria); (3) an exercise or physical activity intervention lasting >2 weeks (RCTs, pre-post, and observational trials); (4) report a measurable exposure variable (i.e., cohort studies: athletes versus non-athletes or sedentary versus active participants, and cross-sectional studies: self-reported or objectively measured habitual

physical activity level); and (5) report CRF-specific outcomes [i.e., absolute ($A\dot{V}O_{2peak}$) or expressed as relative to body mass $\dot{V}O_{2peak}$ ($R\dot{V}O_{2peak}$), evaluated via analysis of expired air during a peak (or symptom-limited) CPET or submaximal prediction exercise test, or PPO].

Studies were excluded if they met the following criteria: (1) non-human; (2) non-original work (i.e., reviews, guideline documents, editorials, viewpoints, letter-to-editor, protocol paper); (3) case reports and case series with a number ($n$) of participants <5 (to increase the robustness of our findings given the inclusion of smaller sample sizes in previous reviews [18,35,36]); (4) non-peer reviewed (i.e., conference proceeding/abstracts/posters); (5) children or adolescents (<18 years); (6) non-SCI (non-injured participants or other neurological conditions); (7) does not report a CRF-specific outcome; (8) single exercise sessions or an intervention <2 weeks; (9) no suitable comparison (i.e., control group or baseline data pre-intervention) or exposure variable measured; and (10) no full text. Studies with concurrent interventions (i.e., diet, lifestyle, or respiratory training) were included only if the effects of exercise could be isolated.

## 2.2. Search strategy

A search of the following electronic databases: MEDLINE (via Pubmed), Excerpta Medica Database (EMBASE; via Ovid), Web of Science, and the Cochrane Central Register of Controlled Trials (CENTRAL) was conducted from their respective inception through to March 25, 2023. Search terms were developed by the corresponding author (TN) and agreed upon by co-authors (AK and MW). The search strategy combined key words describing the following: (1) condition (e.g., SCI); (2) "intervention or exposure variable" (e.g., rehabilitation, exercise, and physical activity); and (3) "outcome" (e.g., $\dot{V}O_{2peak}$ or PPO). Details of the complete search strategy can be found in online Supporting information (Tables A–C in S1 File). Search results were collated using Endnote software (Thomson Reuters, New York, United States of America) and duplicates removed.

## 2.3. Study selection and data extraction

The citations retrieved from the search strategy were screened by title, abstract, and full text by 2 independent reviewers (DH and GB). At each stage of the evaluation, studies were excluded if the inclusion criteria were not satisfied. A conservative approach was taken, whereby if insufficient information was available to warrant study exclusion during the title and abstract stages of screening, studies were retained in the sample for full-text screening. TN resolved any disagreement with regards to study inclusion. There was no restriction on the language of studies. Where necessary, reviewers screened using Google translate [37] or sought assistance from bilingual co-authors.

Two authors (DH and GB) independently extracted data in duplicate using Microsoft Excel. Any disagreements were resolved via mutual consensus. Where more than 1 publication was apparent for the same participants, data were extracted from the study with the largest sample size to avoid duplication. Author, year, study design, sample size, participant demographics/injury characteristics, exercise parameters (including the type, frequency, duration, intensity, and weekly volume), or physical activity exposure details (training history, objective wearable device, or validated self-report questionnaire) and adverse events were extracted. For RCTs, pre-post interventions and observational studies, mean ± standard deviation (SD) for $\dot{V}O_{2peak}$ and PPO outcomes at baseline and post-intervention/control or observation period were extracted to assess change in CRF. For cross-sectional studies, mean ± SD outcomes were extracted for the unique cohorts, along with the significance and magnitude of associations

between CRF and habitual physical activity. Where possible, $\dot{V}O_{2peak}$ values were extracted in relative (mL/kg/min) and absolute (L/min) terms or calculated using pre- and post-intervention body mass values when provided. As it is widely accepted that $\dot{V}O_{2peak}$ should be expressed in relative terms, to account for changes in body mass, results are presented for both $A\dot{V}O_{2peak}$ and $R\dot{V}O_{2peak}$, but the discussion focuses primarily on $R\dot{V}O_{2peak}$. PPO values were extracted in watts (W) only. If there was insufficient information, the authors were contacted via email ($N$ = 13) and given a two-week window to provide additional data or clarity (responses received, $N$ = 8 [38–45]). Detailed notes were recorded outlining the reasons for study inclusion/exclusion and the number of studies included and excluded at each stage.

## 2.4. Risk of bias

Study quality was appraised by at least 2 independent reviewers in duplicate (DH, GB, and SYC), with any conflicts resolved by a third reviewer (TN). The Cochrane Risk of Bias 2 (RoB 2) was used to assess the risk of bias of the RCTs [46]. Reviewers determined the level of bias for each domain using the RoB 2 algorithms and is presented visually using Robvis [47]. Non-randomised designs were assessed using assessment tools generated by the National Institutes of Health (NIH) and National Heart, Lung and Blood Institute (NHLBI, Bethesda, Maryland, USA). Pre-post studies were rated using the Quality Assessment Tool for Before-After (Pre-Post) Studies with No Control Group (12 items) and observational and cross-sectional studies were rated using the Quality Assessment Tool for Observational Cohort and Cross-Sectional Studies (14 items). Studies were subsequently classified as good, fair, or poor quality using the guidance provided within each tool and is presented visually in online Supporting information (S4–S5 and S10–S13 Files).

## 2.5. Data synthesis and analysis

A variety of methods [i.e., indices of heart rate (HR), $\dot{V}O_2$ or ratings of perceived exertion (RPE)] have been utilised in the literature to establish, prescribe, and regulate exercise intensity in the SCI population, which creates complexity when classifying the intensity of exercise. Each intervention was classified as having prescribed either light, moderate, vigorous, or supramaximal-intensity aerobic exercise, based on thresholds proposed by the American College of Sports Medicine (ACSM) [48] (Table A in S2 File). If a study reported a progression in intensity that spanned the moderate and vigorous-intensity categories (e.g., 60% to 65% $\dot{V}O_{2peak}$), it was classified as "moderate-to-vigorous." If insufficient data were provided, studies were classified as "mixed-intensity/cannot determine." Furthermore, where a study reported frequency of sessions or length of interventions as a range (e.g., 6 to 8 weeks), the midpoint was extracted and if a study reported duration as a range (e.g., 40 to 45 min), the greater value was extracted. Descriptions of adverse events in the included studies were also collated. These were categorised into the following subgroups: (1) bone, joint, or muscular pain; (2) autonomic or cardiovascular function; (3) skin irritation or pressure sores; and (4) other.

Means ± SD were estimated from median and interquartile range (IQR) [49] or median and range [50], where required. Where CRF data was only presented in figures, data were extrapolated using Photoshop (Adobe). To combine within-study subgroups and to estimate SD of the delta (Δ) change in CRF using correlation factors, we followed guidance from the Cochrane handbook [49]. Correlation factors were calculated for $A\dot{V}O_{2peak}$, $R\dot{V}O_{2peak}$, and PPO using

studies that reported pre-post SD and SD of the $\Delta$ change using the following equation:

$$Corr = \frac{\left(SD_{pre}\right)^2 + (SD_{Post})^2 - \left(SD_{Change}\right)^2}{2 \times SD_{pre} \times SD_{Post}}.$$

The specific correlation factors that were calculated for each study were averaged across each study design (Table A in S3 File) and applied in the following equation to calculate SD of the change for studies where these values were not reported:

$$SD_{Change} = \sqrt{\left(SD_{pre}\right)^2 + (SD_{Post})^2 - 2 \times corr \times SD_{pre} \times SD_{Post}}$$

where *corr* represents the correlation coefficient.

Since $A\dot{V}O_{2peak}$, $R\dot{V}O_{2peak}$, and PPO are continuous variables, expressed using the same units across studies, we utilised weighted mean differences (WMDs) and 95% confidence intervals (CIs) as summary statistics. A primary meta-analysis was carried out that included RCTs comparing $\Delta$ in CRF outcomes following an exercise intervention relative to control groups. There were not enough studies to perform subgroup analyses within this study design specifically, in accordance with Cochrane recommendations (i.e., a minimum of 10 studies within each subgroup analysis) [51]. Therefore, a pooled meta-analysis describing $\Delta$ in CRF outcomes in response to prospective, well-characterised exercise interventions lasting >2 weeks (e.g., combining exercise intervention-arms from RCTs and pre-post studies) was also conducted to facilitate subgroup comparisons. Nine separate subgroup analyses were performed for this pooled meta-analysis to describe $\Delta$ in each CRF outcome with studies categorised into subgroups based on the following: (1) time since injury [(TSI), e.g., acute (<1-year), chronic (≥1-year)]; (2) neurological level of injury (e.g., tetraplegia, paraplegia); (3) injury severity [e.g., grading in accordance with the American Spinal Injury Association Impairment Scale (AIS): motor-complete (AIS A-B), motor-incomplete (AIS C-D)]; (4) exercise modality [e.g., aerobic volitional upper-body, resistance training, FES, gait training, hybrid/multimodal, behaviour change]; (5) relative exercise intensity (e.g., light, moderate, moderate-to-vigorous, vigorous, supramaximal); (6) method used to prescribe exercise intensity (e.g., $\dot{V}O_2$, HR, RPE, workload); (7) frequency of exercise sessions per week (<3, ≥3 to <5, ≥5); (8) aerobic exercise volume [e.g., SCI-specific exercise guidelines for fitness (40 to 89 min/wk) [21], SCI-specific exercise guidelines for cardiometabolic health (90 to 149 min/wk) [21], achieving general population exercise guidelines (≥150 min/wk) [23]]; and (9) length of intervention (≤6 weeks, >6 to ≤12 weeks, >12 weeks). Studies were also classified as "mixed" (i.e., if cohorts included acute and chronic TSI, tetraplegia and paraplegia, motor-complete and motor-incomplete, prescribed exercise intensity via HR and RPE) or "not reported/cannot determine." Studies classified as "mixed" or "not reported/cannot determine" were not included in the subgroup analyses but are included in the overall WMDs reported herein and the Supporting information (Tables E and F in S5 File). Three additional meta-analyses were conducted for different trial designs: (1) comparing inactive versus active participants (e.g., cross-sectional cohort studies); (2) describing $\Delta$ in CRF outcomes with standard of care inpatient rehabilitation or free-living follow up (e.g., observational studies); and (3) head-to-head comparison of different exercise intensities (RCTs with exercise interventions comparing low or moderate versus vigorous or supramaximal-intensity exercise). Meta-analyses were conducted in R (Version 3.5.1, R Foundation for Statistical Computing, Vienna, Austria) using the package metafor [52]. Statistical heterogeneity was assessed using the $I^2$ and accompanying $p$-value from the chi-squared test. A random effects model was chosen to account for the variability in the true effect

size across studies, given the expected between-study variability of different exercise intervention components and participant characteristics (by nature of SCI being a heterogeneous condition depending on the neurological level and severity of injury). Evidence for differences in effects between the subgroups in the pooled meta-analysis was explored by comparing effects in the subgroups and the corresponding $p$-values for interaction (metagen function from the R package meta [53]). Thresholds for statistically significant subgroup differences were adjusted for the number of subgroup comparisons and individual subgroup $p$-values were adjusted for multiple comparisons via the Bonferroni correction method. To assess the effect of potential outlier studies, leave-one-out analyses were performed with studies removed and pooled WMD recalculated. Sensitivity analyses were also conducted by comparing the WMDs of low and high risk of bias studies, as well as studies with and without imputed data (i.e., extracted from figures or where mean ± SD were calculated from median, IQR, or range), to confirm the robustness of our findings. A further subgroup analysis was performed to compare Δ in CRF outcomes following exercise interventions that matched the CPET modality to the intervention modality (i.e., using an incremental arm-crank ergometry (ACE) CPET to test the effects of an arm-crank exercise intervention). Potential small study effects in the dataset were assessed using funnel plots. Egger's tests were also conducted in R when there was a minimum of 10 studies included in a meta-analysis [51]. Study design statistical power for both the summary effect size and a range of hypothetical effect sizes was calculated and visualised in firepower plots using the metameta R package, recently described by Quintana [54]. Plots were produced for the pre-post exercise interventions alone and for the RCT exercise interventions alone relative to controls (i.e., the primary meta-analysis studies) to facilitate comparisons between study designs. Data is visualised in R (see Github for scripts: https://github.com/jutzca/Exercise-and-fitness-in-SCI). A 2.7 mL/kg/min, and thus 1 metabolic equivalent in SCI (1 SCI-MET) [55], change in $R\dot{V}O_{2peak}$ was considered clinically meaningful.

To explore potential sources of heterogeneity, a random-effects meta-regression was performed using preselected moderator variables in Stata (Version 13, StataCorp LLC, College Station, Texas, USA), adjusted for multiple testing. As per Cochrane recommendations [53], for each included covariate in the model a minimum of 10 studies were required. To achieve this, and to also overcome the issue of collinearity between moderators, some moderators were not included in the analysis. Moderators were selected *a priori*, based on their potential to influence CRF responses. Exercise intensity prescription was later added as a moderator in the meta-regression in light of a recent study challenging strategies for prescribing exercise intensity in individuals with SCI [56]. Moderators fell into 2 categories: *model 1*— participant/injury characteristics [continuous variables: age, TSI, and baseline CRF; categorical variables: sex ($n$ = male), neurological level of injury ($n$ = PARA), severity ($n$ = motor-complete)]; or *model 2*— intervention/study characteristics [continuous variables: exercise session duration, frequency, weekly exercise volume, intervention length; categorical variables: exercise modality, exercise intensity, method of exercise intensity prescription, and risk of bias classification]. Any potential covariates of the effect of $A\dot{V}O_{2peak}$, $R\dot{V}O_{2peak}$, and PPO with $p \leq 0.10$ identified via univariate meta-regression were subsequently included in multivariate meta-regression modelling. The level of significance for multivariate meta-regression was set at $p \leq 0.10$. Because meta-regression can result in inflated false-positive rates when heterogeneity is present, or when there are few studies, a permutation test described by Higgins and Thompson [57] was used to verify the significance of the predictors in the final model, whereby 10,000 permutations were generated.

## 2.6. Certainty on the body of the evidence assessment using the GRADE approach

The Grading of Recommendations Assessment, Development and Evaluation (GRADE) approach [58] was used to evaluate the certainty of the evidence for $\mathrm{A\dot{V}O_{2peak}}$, $\mathrm{R\dot{V}O_{2peak}}$, and PPO for the pooled pre-post and RCT exercise interventions meta-analysis. It was decided that the greater number of studies included in the pooled meta-analysis, in comparison to the primary meta-analysis consisting of RCTs only, would provide a more accurate assessment of the current body of evidence. Two authors (DH and SYC) independently assessed the certainty of evidence for each outcome, with any conflicts resolved by the corresponding author (TN). The certainty of the evidence was graded from "High" to "Moderate," "Low" or "Very Low." GRADE certainty in the evidence was downgraded if one or more of the following criteria were present: (1) risk of bias; (2) inconsistency in the results for a given outcome; (3) indirectness; (4) imprecision; and (5) small study effects.

## 3. Results

The initial database search identified 14,248 articles after removal of duplicates. A further 12,322 studies were removed following the screening of titles and abstracts. The remaining 1,926 articles were selected for full-text review based on inclusion and exclusion criteria (S1 File). Of these, a total of 120 eligible studies, across each specific study design (RCT = 29, pre-post = 67, observational = 6, cross-sectional cohort = 11, cross-sectional association = 7), were included in this review (Fig 1). Twenty-two RCTs were included in the primary meta-analysis (S4 File). Ninety-six studies, comprising the RCTs and pre-post studies (total = 117 independent interventions), were included in the pooled meta-analysis (S5 File). Summaries of the pooled cohorts and descriptions of the individual studies included within each additional meta-analysis are provided as Supporting information (S10–S13 Files).

### 3.1. Primary meta-analysis: RCT exercise intervention versus control groups

Twenty-two RCTs assessed changes in CRF outcomes between exercise intervention ($n = 283$ participants) and control ($n = 252$ participants) groups. Seven RCTs compared changes in CRF outcomes between low-to-moderate and vigorous-to-supramaximal intensity interventions. These specific RCTs are reported in an additional meta-analysis and are therefore not included within the primary meta-analysis. Participant demographics, injury characteristics, exercise intervention parameters, and changes in CRF outcomes for each RCT can be found in Fig 2. A summary of the pooled cohort characteristics, summary statistics for the meta-analysis, and forest plots for each CRF outcome are presented in Supporting information (S4 File). The meta-analysis of RCTs revealed a significantly higher $\mathrm{A\dot{V}O_{2peak}}$ [0.16 (0.07, 0.25) L/min, $p < 0.001$], $\mathrm{R\dot{V}O_{2peak}}$ [2.9 (1.8, 3.9) mL/kg/min, $p < 0.001$], and PPO [9 (5, 14) W, $p < 0.001$] following exercise interventions relative to controls. There was significant heterogeneity present across all CRF outcomes ($p < 0.001$), with $I^2$ values of 87%, 93%, and 88% for $\mathrm{A\dot{V}O_{2peak}}$, $\mathrm{R\dot{V}O_{2peak}}$, and PPO, respectively.

### 3.2. Pooled meta-analysis: Pre-post and RCT exercise interventions

CRF responses were pooled across 96 studies, comprising 117 exercise interventions in total, taken from 81 pre-post exercise interventions and 36 independent exercise intervention arms from RCTs. Some studies included multiple exercise intervention arms/phases; hence, the greater total number of exercise interventions than studies. A summary of the demographic/

PRISMA 2020 flow diagram for new systematic reviews which included searches of databases and registers only

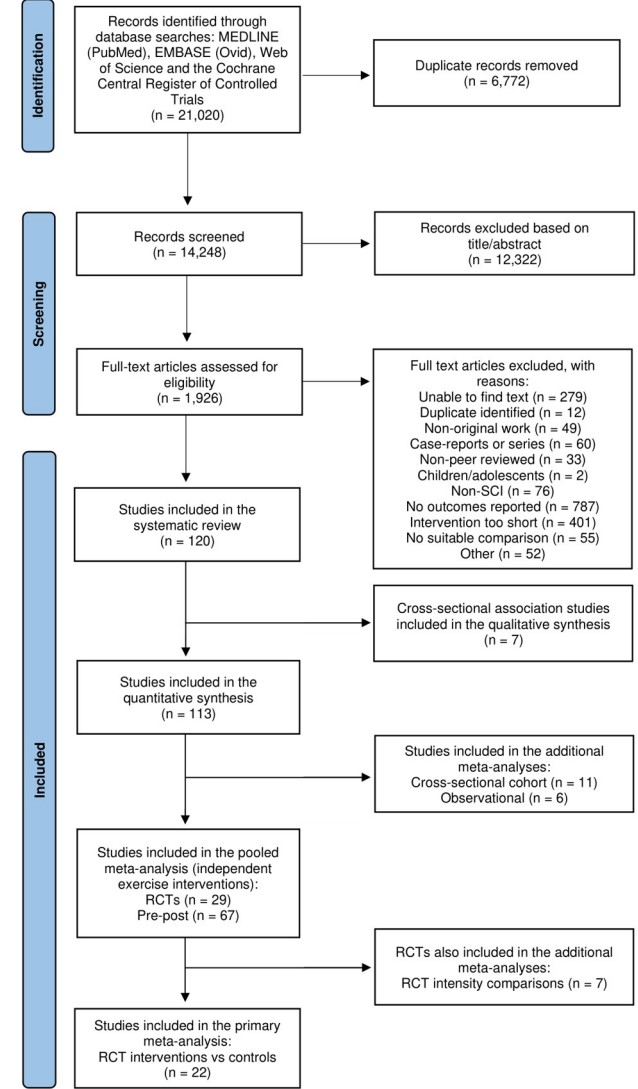

**Fig 1. PRISMA flow diagram.** PRISMA, Preferred Reporting Items for Systematic Reviews and Meta-Analyses.

injury characteristics and intervention parameters for the pooled cohort (i.e., all 117 interventions) included in the analyses for $A\dot{V}O_{2peak}$, $R\dot{V}O_{2peak}$, and PPO are presented in Supporting information (Tables A and B in S5 File). Participant demographics and exercise intervention parameters across the total 117 exercise interventions are summarised below. Summaries of the interventions for each CRF outcome, along with details of the specific studies, are presented in online Supporting information (S5 File).

**3.2.1. Participants.** Across the 117 exercise interventions, there were a total of 1,331 participants. Most interventions included both males and females (63% of studies), where females made up between 6% and 80% of the mixed cohorts. There were no female-only cohorts. Mean age ranged between 23 and 58 years and the majority of participants had chronic injuries (64% >1-year), with mean TSI ranging between 56 days to 24 years. Sixty-three interventions included a mixed cohort of paraplegia and tetraplegia, of which individuals with paraplegia

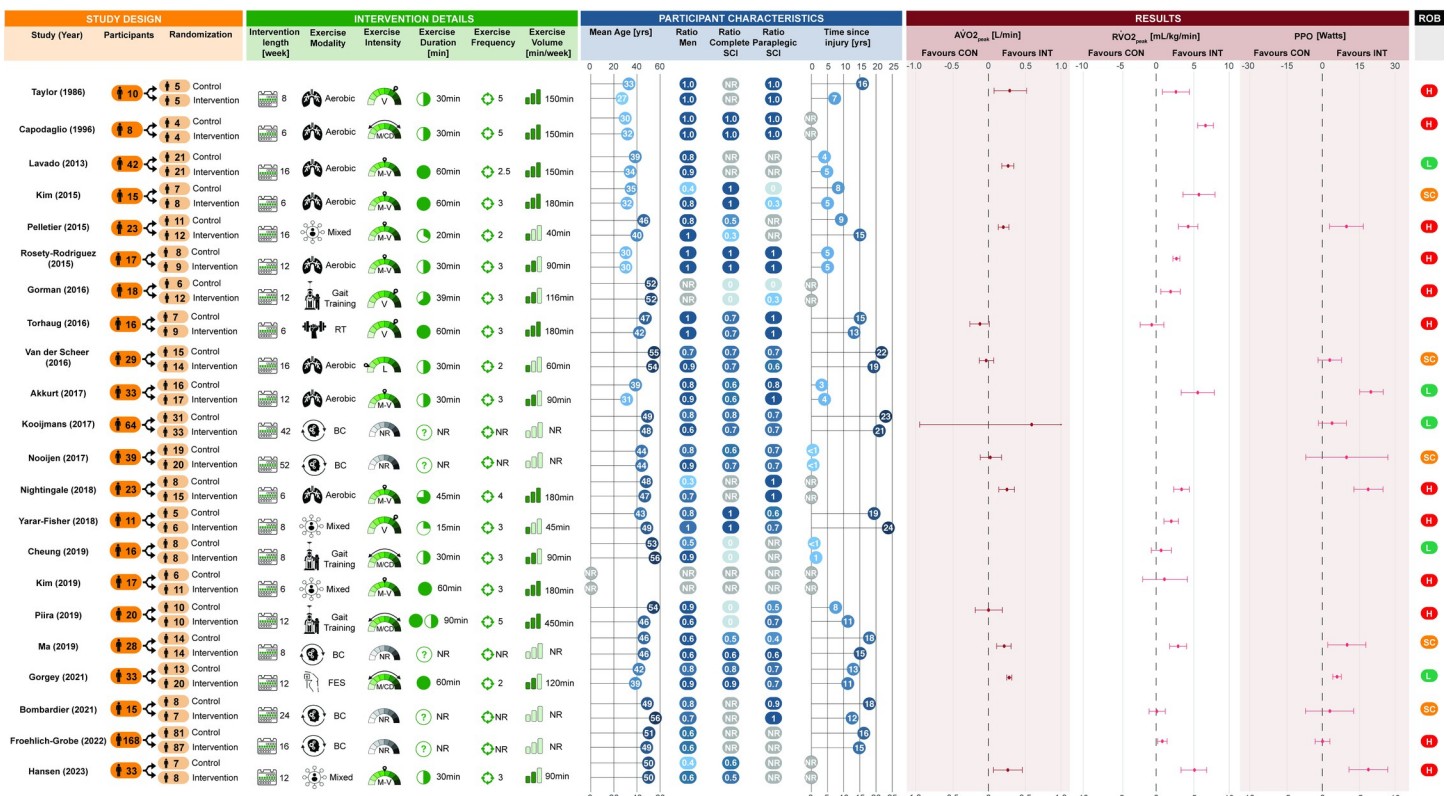

**Fig 2. Summary of the study design, exercise intervention parameters, participant demographics/injury characteristics, and changes in cardiorespiratory fitness outcomes for each RCT included in the primary meta-analysis.** Studies were assessed for risk of bias (ROB) and rated as either low (L), some concerns (SC) or high (H) risk. AV̇O₂peak, absolute peak oxygen uptake; BC, behaviour change; FES, functional electrical stimulation; L, light-intensity; M, moderate-intensity; M/CD, mixed/cannot determine; M-V, moderate-to-vigorous-intensity; NR, not reported; PPO, peak power output; RT, resistance training; RV̇O₂peak, relative peak oxygen uptake; SCI, spinal cord injury; V, vigorous-intensity.

made up between 10% and 88% of the mixed cohorts. Four interventions recruited individuals with tetraplegia-only, 39 paraplegia-only, and 11 did not specify. Participants across all AIS groups were included, of which 41 interventions were motor-complete-only, 19 were motor-incomplete-only, and 20 did not report. Thirty-seven interventions recruited both motor-complete and incomplete individuals, of which 33% were motor-incomplete. Weighted mean AV̇O₂peak and RV̇O₂peak at baseline was 1.28 (0.51 to 3.50) L/min and 17.8 (7.3 to 36.9) mL/kg/min, respectively, and PPO was 49 (0 to 168) W.

**3.2.2. Exercise intervention characteristics.** Length of interventions ranged from 4 to 52 weeks, and while most studies reported a specific, predetermined intervention length, some reported a range [59–61], a total or targeted number of sessions [60,62–66], or provided an average [65,67,68]. Exercise sessions were completed between 2 and 7 times per week. Eleven studies reported a range (e.g., "two to three sessions") or maximum frequency (e.g., "up to three sessions/week") [60,63,66,69–76], and frequency was either not reported or could not be determined in 7 studies [39,77–82]. The remainder reported an exact frequency (e.g., 3 sessions per week). The duration of exercise sessions ranged from 5 to 90 min, with 5 studies reporting a range (e.g., 20 to 30 min) [60,76,83–85] and 6 studies reporting a progression to a target duration [63,86–90]. Duration was not reported or could not be determined in 15 studies. Based on current exercise guidelines, 23 interventions prescribed exercise within the SCI-specific exercise guidelines for fitness (40 to 89 min/week), 45 interventions targeted the SCI-

specific exercise guidelines for cardiometabolic health (90 to 149 min/week), and 30 were greater than general population exercise guidelines ($\geq$150 min/week).

Forty-two interventions utilised aerobic upper-body exercise, 7 upper-body resistance training, 22 FES, 15 gait training, 5 behaviour change, and 26 mixed/multimodal interventions. Following the ACSM thresholds, 1 intervention prescribed light-intensity (1%), 17 prescribed moderate-intensity (14%), 35 prescribed moderate-to-vigorous-intensity (30%), 26 prescribed vigorous-intensity (22%), and 2 prescribed supramaximal-intensity exercise (2%). Intensity could not be determined from 36 interventions (31%). With regards to exercise intensity prescription methods, 35 interventions used HR, regulated either via $HR_{peak}$ (% $HR_{peak}$, i.e., determined via a CPET; $N = 10$), $HR_{max}$ (%$HR_{max}$, i.e., age-predicted; $N = 10$), or HR reserve (%HRR; $N = 15$). Fourteen interventions established intensity using $\dot{V}O_{2peak}$ (%$\dot{V}O_{2peak}$; $N = 13$) or $\dot{V}O_2$ reserve (%$\dot{V}O_{2reserve}$; $N = 1$) calculated from the pre-intervention CPET. Fourteen interventions utilised RPE, using either the Borg CR10 scale ($N = 7$) or the Borg 6–20 scale ($N = 7$). Workload was used to prescribe intensity in 11 interventions, via a percentage of PPO (%PPO; $N = 6$), 1 repetition maximum (%1RM; $N = 4$), or maximal tolerated power (%MTP; $N = 1$). Forty-three interventions either used a mixture of prescription methods or intensity could not be classified.

**3.2.3. Adverse events.** Adverse events were described in 18 interventions, comprising at least 49/1,331 (3.7%) participants (S6 File). These events were related to: (1) bone, joint, or muscular pain ($n = 10$ participants); (2) autonomic or cardiovascular function ($n = 8$ participants); (3) skin irritation or pressure sores ($n = 18$ participants); and (4) other events including anxiety, nausea, dizziness, and issues with testing equipment ($n = 3$ participants). The following adverse events were reported in 4 other pre-post studies but could not be categorised as above. Beillot and colleagues [77] stated that participants experienced "spontaneous fractures of lower limbs, occurrence of a syringomyelia and pressure sores at the foot and ankle" ($n = 10$), but did not define the number of participants who sustained each event. Likewise, Janssen and Pringle [70] reported "lightheadedness in some subjects" and Gibbons and colleagues [91] stated that "a number of participants showed some level of autonomic dysreflexia during the FES response test", but both studies did not quantify further. Vestergaard and colleagues [92] reported adverse events relating to "slight non-persisting pain in neck ($n = 1$), arms and shoulders ($n = 4$) during and between training sessions, dizziness that disappeared after 5 min ($n = 1$), feeling tired in the head/dizziness that disappeared after training with no other signs of autonomic hyperreflexia ($n = 2$), increased spasms ($n = 2$), and vomiting just after training ($n = 2$)." However, as only 7 participants completed the intervention, it cannot be determined whether these events were reported for 1 or multiple participants.

**3.2.4. Change in CRF outcomes.** Seventy-four exercise interventions assessed the change in $A\dot{V}O_{2peak}$, revealing a significant increase of 0.22 [0.17, 0.26] L/min ($p < 0.001$). There were no significant subgroup differences for any of the 9 subgroup analyses. Seventy-nine exercise interventions assessed the change in $R\dot{V}O_{2peak}$, revealing a significant increase of 2.8 [2.3, 3.3] mL/kg/min ($p < 0.001$). There were significant subgroup differences for exercise modality ($p = 0.002$) and length of intervention ($p = 0.01$), but there were no other differences (Fig 3). Sixty-five exercise interventions assessed the change in PPO, revealing a significant increase of 11 [9,13] W ($p < 0.001$). There were significant subgroup differences for TSI ($p = 0.01$), neurological level of injury ($p < 0.01$), exercise modality ($p = 0.002$), and frequency ($p < 0.001$) (Fig 3). There was significant heterogeneity present across all CRF outcomes ($p < 0.001$), with $I^2$ values of 72%, 53%, and 78% for $A\dot{V}O_{2peak}$, $R\dot{V}O_{2peak}$, and PPO, respectively. Forest plots for each subgroup analysis are presented in Supporting information (S5 File).

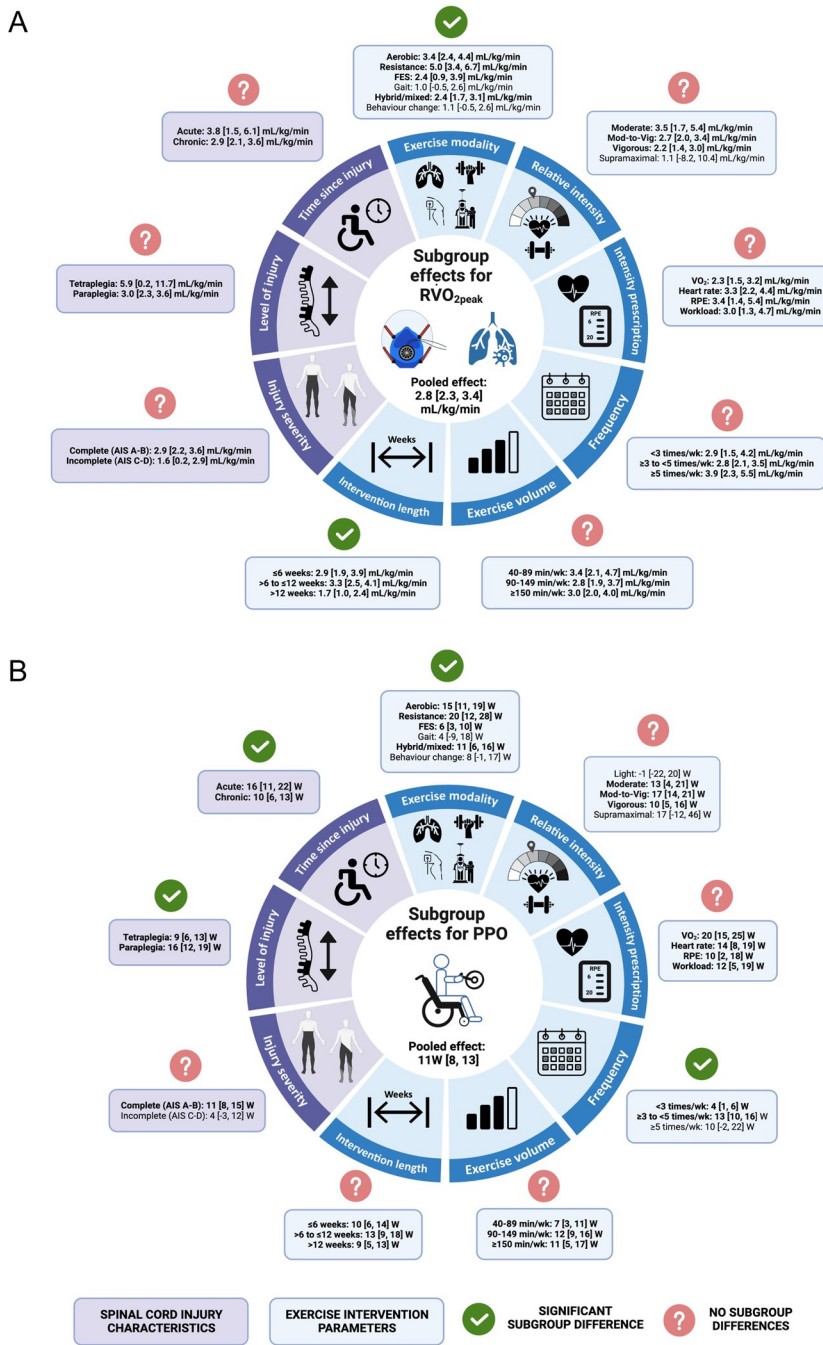

**Fig 3.** Overview of the subgroup effects for relative peak oxygen uptake (A; RVO$_{2peak}$) and peak power output (B; PPO) for pooled pre-post and RCT exercise interventions based on injury-specific characteristics (purple) and exercise intervention parameters (blue). Individual subgroup effects are highlighted in bold. Significant differences within each subgroup category are represented by green ticks, whereas subgroup categories without significant differences are represented by red question marks. AIS, American Spinal Injury Association Impairment Scale; FES, functional electrical stimulation; Mod-to-Vig, moderate-to-vigorous; NR/CD, not reported/cannot determine; RPE, rating of perceived exertion; VO$_2$, volume of oxygen consumption W, watts.

An additional subgroup analysis grouped interventions into those that matched the CPET modality to the exercise intervention and those that did not. Following the adjustment for subgroup comparisons, there were no significant differences in $A\dot{V}O_{2peak}$, $R\dot{V}O_{2peak}$, or PPO (S8 File). However, there were trends for a greater $A\dot{V}O_{2peak}$ ($p = 0.05$) and $R\dot{V}O_{2peak}$ ($p = 0.06$) in studies with matched CPET and intervention modalities in comparison to interventions without matched modalities. A sub-analysis for gait training interventions alone also revealed no subgroup differences between studies that used upper-body or treadmill CPETs to determine CRF outcomes (S9 File).

**3.2.5. Sensitivity analyses.** Leave-one-out analyses identified outliers for $A\dot{V}O_{2peak}$ [42,61,93] (S7 File). A sensitivity analysis for risk of bias revealed no differences in the pooled effects for low and high risk of bias studies (S7 File). A sensitivity analysis comparing studies with imputed data via conversion of medians (IQR), extrapolated data from figures, and studies without imputed data revealed no differences in the pooled effects for any outcome (S7 File).

**3.2.6. Study-design statistical power considerations.** Median statistical power and observed effect sizes for each CRF outcome and study design (i.e., pre-post studies compared to RCT interventions relative to controls) are reported in Fig 4. Across each CRF outcome, median statistical power was higher for the RCTs included in the primary meta-analysis in comparison to the pre-post studies. In general, this indicates that the RCT studies were designed to reliably detect a wider range of effect sizes in comparison to the pre-post studies.

**3.2.7. Meta-regression.** *Model 1—Participant and injury characteristics.* Exercise interventions with a greater mean age of participants were associated with smaller changes in the effect estimates for $A\dot{V}O_{2peak}$ ($p = 0.08$) and $R\dot{V}O_{2peak}$ ($p = 0.01$). The coefficients indicate that for every one-year increase in mean age of participants in an exercise intervention, the effect on $A\dot{V}O_{2peak}$ and $R\dot{V}O_{2peak}$ decreases on min and 0.041/min and 0.041 mL/kg/min, respectively, holding all other covariates constant (Table 1). There were no associations between the other moderator variables included in this model and $\Delta A\dot{V}O_{2peak}$ or $\Delta R\dot{V}O_{2peak}$. While there were no significant associations between $\Delta$PPO and the other moderator variables (Table 1), there was a trend for an association between PPO and TSI ($p = 0.18$). The coefficient for TSI indicates that for every additional year since injury, the effect on PPO following an exercise intervention decreases by 1.5W on average (Table 1).

*Model 2—Exercise intervention and study characteristics.* There were no significant associations between the exercise intervention and study characteristics included in model 2 for $\Delta A\dot{V}O_{2peak}$, $\Delta R\dot{V}O_{2peak}$, or $\Delta$PPO (Table 1).

**3.2.8. Small study effects.** Egger's tests for funnel plot asymmetry were not statistically significant for $A\dot{V}O_{2peak}$ ($Z = -1.20$, $p = 0.23$), $R\dot{V}O_{2peak}$ ($Z = -0.44$, $p = 0.66$), or PPO ($Z = 0.76$, $p = 0.45$). Funnel plots are provided in Supporting information (S5 File).

## 3.3. Additional meta-analyses

**3.3.1. Cross-sectional studies.** Eleven studies included cross-sectional data comparing CRF outcomes in active ($n = 182$ participants) versus inactive ($n = 134$ participants) individuals with SCI. Inactive participants were mainly classified as sedentary, whereas active participants varied from recreationally active wheelchair sport players to paralympic athletes. A meta-analysis of cross-sectional cohort studies revealed significantly ($p < 0.001$) higher $A\dot{V}O_{2peak}$ [0.55 (0.43, 0.67) L/min], $R\dot{V}O_{2peak}$, [9.1 (7.0, 11.2) mL/kg/min], and PPO [38 (32, 45) W] in active compared to inactive individuals with SCI (S10 File). Given the significant heterogeneity in $R\dot{V}O_{2peak}$, a sensitivity analysis was conducted to compare inactive individuals with either "active" or "elite athletes." There was a significantly higher $R\dot{V}O_{2peak}$

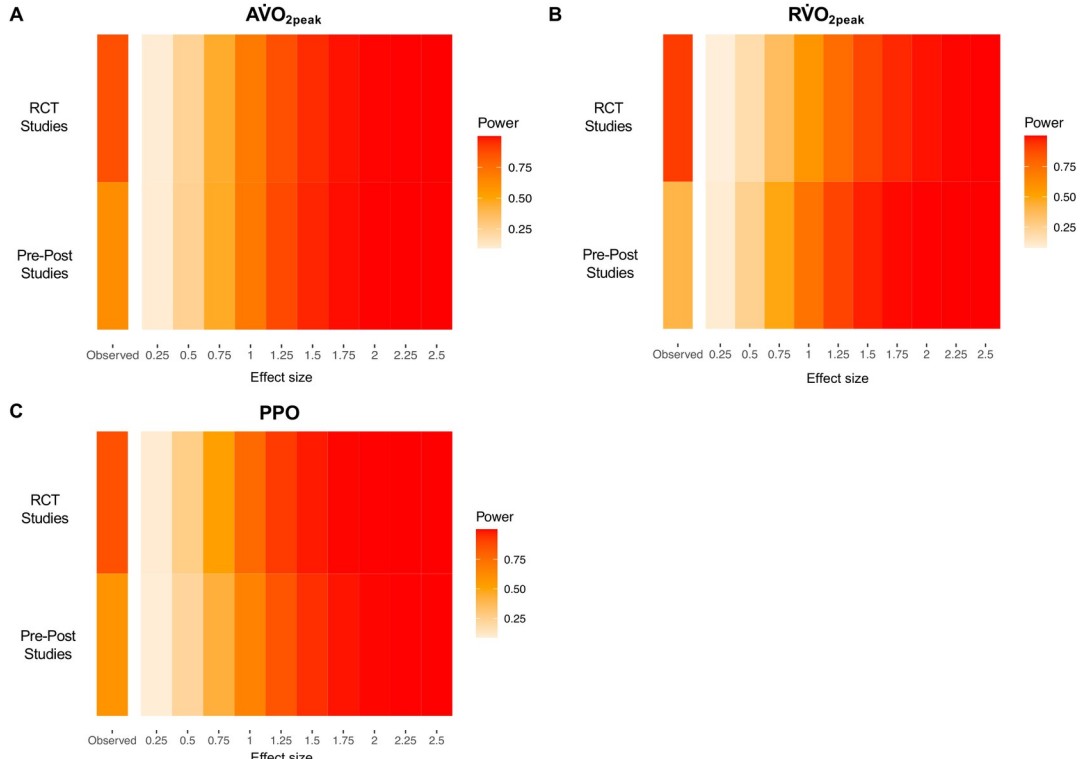

**Fig 4. Firepower plots visualising study-level statistical power for a range of effect sizes for pre-post studies alone in comparison to RCT studies included in the primary meta-analysis (i.e., exercise interventions relative to control groups) for each cardiorespiratory fitness outcome.** Hedges' g effect sizes were calculated for each study design and used as the observed effect size, as described by Quintana [54]. (A) Pre-post studies describing changes in absolute peak oxygen uptake (A$\dot{V}O_{2peak}$) had a median power of 62% for detecting an effect size of 0.90, whereas the RCTs had a median power of 86% for detecting an effect size of 1.25. (B) Pre-post studies describing changes in relative peak oxygen uptake (R$\dot{V}O_{2peak}$) had a median power of 41% for detecting an effect size of 0.67, whereas the RCTs had a median power of 91% for detecting an effect size of 1.57. (C) Pre-post studies describing changes in PPO had a median power of 58% for detecting an effect size of 0.91, whereas the RCTs had a median power of 86% for detecting an effect size of 1.13. PPO, peak power output; RCT, randomised-controlled trial.

[6.4 (4.7, 8.1) mL/kg/min, $p < 0.001$] in "active" compared to inactive individuals, but an even higher R$\dot{V}O_{2peak}$ [11.2 (9.6, 12.9) mL/kg/min, $p < 0.001$] in "elite athletes" compared to inactive (S10 File).

Seven studies ($n$ = 581 participants) included cross-sectional data and assessed associations between habitual physical activity level (as a continuous variable) and CRF outcomes (S11 File). Six studies assessed physical activity exposure using self-report methods [38,40,94–97], whereas 1 study used a validated research-grade wearable device [98]. The measurement period used to capture physical activity dimensions ranged from 3 to 7 days. There was considerable variability across studies with regards to the physical activity dimensions captured: hours per week of exercise/sport, minutes per day or week of mild, moderate, heavy-intensity for the subcategories of leisure time physical activity (LTPA), lifestyle or household activity or cumulative activity. Collectively, data indicates significant positive correlations of a larger magnitude between CRF/PPO outcomes and the volume of sport, exercise or LTPA rather than household activity. The only study to use a validated wearable device indicated that participants performing ≥150 min/week of moderate-to-vigorous physical activity (MVPA) had a significantly higher CRF relative to a low activity group (performing <40 min/wk). Whereas,

**Table 1. Meta-regression models with adjusted values for each CRF outcome.**

| Covariate | Coef. | Std.Err. | t | Unadjusted P>t | 95% CI | Adjusted P>t |
|---|---|---|---|---|---|---|
| AVO$_{2peak}$ Model 1 ($N = 74$)[1] (#covariates = 5) | | | | | | |
| Male | 0.005 | 0.003 | 1.530 | 0.131 | −0.002, 0.012 | 0.448 |
| Mean age | −0.003 | 0.001 | −2.430 | 0.018 | −0.005, −0.001 | **0.083** |
| TSI | −0.000 | 0.001 | −0.440 | 0.665 | −0.003, 0.002 | 0.995 |
| Injury class | 0.003 | 0.003 | 0.850 | 0.398 | −0.004, 0.009 | 0.899 |
| Severity | −0.007 | 0.004 | −1.910 | 0.061 | −0.015, 0.000 | 0.219 |
| AVO$_{2peak}$ Model 2 ($N = 74$)[2] (#covariates = 6) | | | | | | |
| Exercise type | −0.009 | 0.014 | −0.700 | 0.489 | −0.037, 0.018 | 0.967 |
| Exercise intensity | −0.010 | 0.015 | −0.710 | 0.481 | −0.039, 0.019 | 0.972 |
| Intervention length | −0.000 | 0.002 | −0.130 | 0.894 | −0.004, 0.003 | 1.000 |
| Minutes | 0.004 | 0.008 | 0.490 | 0.626 | −0.012, 0.020 | 0.994 |
| Frequency | −0.012 | 0.011 | −1.110 | 0.269 | −0.035, 0.010 | 0.831 |
| Volume | 0.002 | 0.003 | 0.790 | 0.434 | −0.003, 0.007 | 0.937 |
| RVO$_{2peak}$ Model 1 ($N = 79$)[3] (#covariates = 5) | | | | | | |
| Male | 0.060 | 0.044 | 1.370 | 0.174 | −0.027, 0.146 | 0.651 |
| Mean age | −0.041 | 0.013 | −3.200 | 0.002 | −0.066, −0.015 | **0.013** |
| TSI | −0.005 | 0.012 | −0.430 | 0.666 | −0.030, 0.019 | 0.996 |
| Injury class | −0.031 | 0.047 | −0.650 | 0.518 | −0.125, 0.064 | 0.976 |
| Severity | −0.044 | 0.040 | −1.110 | 0.272 | −0.124, 0.035 | 0.819 |
| RVO$_{2peak}$ Model 2 ($N = 79$)[4] (#covariates = 6) | | | | | | |
| Exercise type | −0.229 | 0.145 | −1.580 | 0.120 | −0.518, 0.061 | 0.497 |
| Exercise intensity | −0.093 | 0.189 | −0.490 | 0.622 | −0.470, 0.283 | 0.996 |
| Intervention length | −0.028 | 0.033 | −0.870 | 0.385 | −0.093, 0.036 | 0.930 |
| Risk of bias | −0.034 | 0.329 | −0.100 | 0.919 | −0.690, 0.623 | 1.000 |
| Minutes | −0.023 | 0.057 | −0.400 | 0.687 | −0.137, 0.091 | 0.998 |
| Frequency | 0.183 | 0.141 | 1.290 | 0.200 | −0.099, 0.464 | 0.705 |
| PPO Model 1 ($N = 65$)[5] (#covariates = 5) | | | | | | |
| Male | −0.137 | 0.210 | −0.650 | 0.518 | −0.557, 0.284 | 0.972 |
| Mean age | −0.057 | 0.073 | −0.780 | 0.439 | −0.204, 0.090 | 0.940 |
| TSI | −0.153 | 0.069 | −2.220 | 0.031 | −0.291, −0.015 | 0.177 |
| Injury class | 0.005 | 0.173 | 0.030 | 0.978 | −0.342, 0.351 | 1.000 |
| Severity | −0.065 | 0.173 | −0.380 | 0.707 | −0.410, 0.280 | 0.997 |
| PPO Model 2 ($N = 65$)[6] (#covariates = 5) | | | | | | |
| Exercise type | −0.172 | 0.687 | −0.250 | 0.803 | −1.547, 1.203 | 0.999 |
| Exercise intensity | −1.310 | 0.724 | −1.810 | 0.076 | −2.759, 0.140 | 0.202 |
| Minutes | −0.064 | 0.314 | −0.200 | 0.840 | −0.693, 0.566 | 0.999 |
| Frequency | 0.288 | 0.687 | 0.420 | 0.676 | −1.087, 1.664 | 0.989 |
| Volume | −0.176 | 0.159 | −1.110 | 0.272 | −0.495, 0.142 | 0.669 |

Permutations = 10,000

[1] Adj R-squared = 14.49%; Model F(5,68) = 3.10; Prob > F = 0.01

[2] Adj R-squared = −6.11%; Model F(6,67) = 0.50; Prob > F = 0.80

[3] Adj R-squared = 10.57%; Model F(5,73) = 2.76; Prob > F = 0.02

[4] Adj R-squared = 5.0%; Model F(6,72) = 1.60; Prob > F = 0.16

[5] Adj R-squared = 2.98%; Model F(5,59) = 1.41; Prob > F = 0.23

[6] Adj R-squared = 0.23%; Model F(7,59) = 1.02; Prob > F = 0.41

Significance is highlighted in bold. Adj R$^2$, proportion of between-study variance explained; AVO$_{2peak}$, absolute peak oxygen consumption; Coef, coefficient of variation; Model F, joint test for all covariates; Prob > F, with Knapp–Hartung modification; RVO$_{2peak}$, relative peak oxygen consumption; Std.Err, standard error; TSI, time since injury.

there was no significant difference in CRF between the low activity group and participants achieving the SCI-specific exercise guidelines (40 to 149 min/week) [98]. These data have been replicated in a recent study, which compared CRF outcomes for participants achieving either the SCI-specific exercise guidelines for fitness (40 to 89 min/week) and cardiometabolic health (90 to 149 min/week) [21] with the Exercise and Sport Science Australia (ESSA) position statement recommending that individuals with SCI achieve ≥150 min/week [24]. There were no differences in CRF outcomes between individuals classified as inactive and those meeting the current SCI-specific exercise guidelines, yet there were significant differences across all CRF outcomes between inactive individuals and those achieving the ESSA guidelines. Furthermore, A$\dot{V}O_{2peak}$ and PPO were greater for individuals meeting the ESSA guidelines, relative to the current SCI-specific exercise guidelines. Across studies, significant, positive correlations were reported for the amount of moderate-to-vigorous LTPA or cumulative activity with CRF/PPO outcomes, which was not the case for mild or light-intensity activity.

**3.3.2. Observational inpatient rehabilitation or community free-living studies.** Six studies ($n$ = 354 participants) included observational longitudinal data and assessed changes in CRF outcomes following either standard of care inpatient rehabilitation [99–101] or a period of community free-living [99,102,103]. While 1 study described a training programme for elite-level wheelchair rugby players, it could not be determined whether players adhered to the prespecified training programme throughout the season, and therefore, the study was categorised as a community free-living observational study in this review [104]. The duration between assessments for standard of care varied, ranging from 5 to 28 weeks, whereas the follow-up period for community observations ranged from 30 weeks to 2.9 years. Reporting on the therapies used within standard of care was poor and only 1 study included a measurement of physical activity during the community-based free-living follow-up (self-reported mean sport activity) [102]. There were significant improvements following standard of care, but not following community-based free-living, in A$\dot{V}O_{2peak}$ [0.12 (0.07, 0.17) L/min, $p < 0.001$ versus 0.09 (0.00, 0.19) L/min, $p = 0.06$] and R$\dot{V}O_{2peak}$ [2.1 (1.0, 3.2) mL/kg/min, $p < 0.001$ versus −0.5 (−2.5, 1.5) mL/kg/min, $p = 0.64$] (S12 File). Significant improvements in PPO were identified following both standard of care [6 (3, 9) W, $p < 0.001$] and community-based free-living [7 (2, 12) W, $p = 0.006$] (S12 File).

**3.3.3. Exercise intensity comparisons from RCTs.** Seven RCTs compared changes in CRF outcomes between low and moderate ($n$ = 52 participants) and vigorous or supramaximal ($n$ = 51 participants) exercise intensity groups. These studies utilised upper-body aerobic exercise and gait training. A meta-analysis revealed no significant differences between low or moderate and vigorous or supramaximal intensity in A$\dot{V}O_{2peak}$ ($p = 0.67$), R$\dot{V}O_{2peak}$ ($p = 0.88$), or PPO ($p = 0.62$) (S13 File).

## 3.4. Risk of bias

Full risk of bias assessments for pre-post and RCT interventions can be found in Supporting information (S4 and S5 and S13 Files). Twenty-six pre-post studies were rated as having good, 27 as having fair, and 14 as having poor methodological quality. Six RCTs were rated as having a low risk of bias, 8 as having some concerns, and 15 as having a high risk of bias. The most common domains in the RCTs with either some concerns or high risk were "bias in the measurement of the outcome" and "bias in selection of the reported result." Reporting was inadequate in many of the included studies, which made the assessment of risk of bias challenging. Notably, reporting of blinding, eligibility, or selection criteria, as well as the enrolment of participants (i.e., a lack of CONSORT flow diagrams) was poor. Individual risk of bias assessments for every study design are provided in Supporting information (S4 and S5 and S10–S13 Files).

### 3.5. Evidence appraisal using GRADE

Overall, the GRADE assessment revealed a "Moderate" certainty in the body of evidence for improvements in $R\dot{V}O_{2peak}$, but a "Low" certainty in the body of evidence for improvements in $A\dot{V}O_{2peak}$ and PPO (Table 2). The certainty rating for $A\dot{V}O_{2peak}$ was downgraded due to imprecision and a lack of high-quality study designs, whereas $R\dot{V}O_{2peak}$ was downgraded as a result of imprecision. The certainty rating for PPO was downgraded due to imprecision and inconsistency, resulting from considerable heterogeneity in the included exercise interventions.

## 4. Discussion

This review provides a large evidence-based summary and appraisal of prescribed and prospective exercise interventions >2 weeks and changes in CRF in individuals with SCI. The results from the primary meta-analysis of RCTs support the role of exercise in improving CRF in this population by 0.16 L/min and 9W in $A\dot{V}O_{2peak}$ and PPO, respectively. The primary meta-analysis also indicates a clinically meaningful change in $R\dot{V}O_{2peak}$ of 2.9 mL/kg/min. Subgroup analyses in the pooled meta-analysis revealed no differences in $A\dot{V}O_{2peak}$ with respect to injury characteristics and exercise intervention parameters. However, there were differences in $R\dot{V}O_{2peak}$ based on exercise modality and length of intervention. There were also significant subgroup differences for PPO based on TSI, neurological level of injury, exercise modality, and frequency of sessions per week. The GRADE assessment including RCTs and pre-post intervention studies revealed "Moderate" certainty in the evidence for improvements in $R\dot{V}O_{2peak}$, yet "Low" certainty in the evidence for significant improvements in $A\dot{V}O_{2peak}$ and PPO.

Following exercise interventions, $\dot{V}O_{2peak}$ increases in individuals with both acute and chronic SCI. However, this review highlights the need for more exercise interventions in the acute phase post-SCI. Indeed, a recent review by Van der Scheer and colleagues [35] rated the confidence in the evidence base for exercise in acute SCI as "Very Low," and called for more RCTs to control for the deteriorations in fitness and health occurring almost immediately following SCI. With regards to PPO in the current review, the meta-regression revealed a trend for an association between TSI and changes in PPO, with individuals with long-term injuries exhibiting smaller changes in PPO in comparison to those with relatively newer injuries. The subgroup analysis based on TSI also revealed that individuals with acute SCI exhibit a greater change than individuals with chronic SCI. This may be due to exercise being delivered in combination with standard of care inpatient rehabilitation for individuals with acute injuries, representing an additive effect. Indeed, the additional meta-analysis with longitudinal observational studies indicates a 6W improvement in PPO with standard of care inpatient rehabilitation alone during the subacute period. This could be influenced by spontaneous motor recovery in the first few months following SCI [105]. Or alternatively, this subacute period may provide a familiarisation effect to novel modalities of exercise that patients were previously unaccustomed to and/or provides a notable stimulus to naive upper-body musculature following a period of bed rest and deconditioning, possibly representing a regression to the mean artefact. Ultimately, more rigorous RCTs are required in the subacute phase post-SCI that compare standard of care versus standard of care plus a specific exercise intervention to truly quantify the independent effects of a prescribed exercise intervention in the inpatient setting.

Exercise results in improved $\dot{V}O_{2peak}$ regardless of the neurological level of injury. In particular, this review reveals a pooled improvement of 5.9 mL/kg/min in studies that included only

**Table 2. Grading of recommendations assessment, development, and evaluation analysis for each CRF outcome.**

| | A$\dot{V}O_{2peak}$ (L/min) | R$\dot{V}O_{2peak}$ (mL/kg/min) | PPO (W) |
|---|---|---|---|
| Summary of findings according to GRADE analysis | | | |
| GRADE | LOW | MODERATE | LOW |
| Comments | Our confidence in the effect estimate is limited: The true effect may be substantially different from the estimate of the effect. Study design and imprecision reduced the GRADE to Low. The evidence supporting improvements in A$\dot{V}O_{2peak}$ is predominantly in young and middle-aged males that had been injured for >1-year (chronic TSI). Participants were mostly paraplegic (67%) but there were a mixture of injury severities (AIS A-D). There were no subgroup differences to suggest optimal training parameters. | We are moderately confident in the effect estimate: The true effect is likely to be close to the estimate of the effect, but there is a possibility that it is substantially different. Imprecision reduced the GRADE to Moderate. The evidence supporting improvements in R$\dot{V}O_{2peak}$ is predominantly in young and middle-aged males that had been injured for >1-year (chronic TSI). Participants were mostly paraplegic (69%) but there were a mixture of injury severities (AIS A-D). Please see Fig 3 for subgroup effects. | Our confidence in the effect estimate is limited: The true effect may be substantially different from the estimate of the effect. Inconsistency and imprecision reduced the Grade to Low. The evidence supporting improvements in PPO is predominantly in young and middle-aged males that had been injured for >1-year (chronic TSI). Participants were mostly paraplegic (73.5%) but there were a mixture of injury severities (AIS A-D). Please see Fig 3 for subgroup effects. |
| Lower quality criteria | | | |
| Study design | Mixture of RCTs and pre-post studies with no control groups. Overall WMDs for RCT interventions relative to controls and pre-post interventions only: RCTs (0.16 L/min) and pre-post studies (0.23 L/min). DOWNGRADE | Mixture of RCTs and pre-post studies with no control groups. Overall WMDs for RCT interventions relative to controls and pre-post interventions only: RCTs (2.9 mL/kg/min) and pre-post studies (2.9 mL/kg/min). NO DOWNGRADE | Mixture of RCTs and pre-post studies with no control groups. Overall WMDs for RCT interventions relative to controls and pre-post interventions only: RCTs (9 W) and pre-post studies (11 W). NO DOWNGRADE |
| Risk of bias (RoB) | Sensitivity analysis revealed no significant difference between studies with low and high risk of bias. NO DOWNGRADE | Sensitivity analysis revealed no significant difference between studies with low and high risk of bias. NO DOWNGRADE | Sensitivity analysis revealed no significant difference between studies with low and high risk of bias. NO DOWNGRADE |
| Inconsistency of results | Effect estimates were consistent, with 89% of the included exercise interventions favouring an increase in A$\dot{V}O_{2peak}$, but most had a low effect estimate. Large overlap in confidence intervals. $I^2$ = 72% NO DOWNGRADE | Effect estimates were consistent, with 92% of the included exercise interventions favouring an increase in R$\dot{V}O_{2peak}$, and most had a large effect estimate. Large overlap in confidence intervals. $I^2$ = 53% NO DOWNGRADE | Effect estimates were consistent, with 94% of the included exercise interventions favouring an increase in PPO, and most had a large effect estimate. Large overlap in confidence intervals. $I^2$ = 78% DOWNGRADE |
| Indirectness | Most studies (81%) included A$\dot{V}O_{2peak}$ in their main outcome measures, across a range of participant characteristics. NO DOWNGRADE | Most studies (68%) included R$\dot{V}O_{2peak}$ in their main outcome measures, across a range of participant characteristics. NO DOWNGRADE | Most studies (80%) included PPO in their main outcome measures, across a range of participant characteristics. NO DOWNGRADE |
| Imprecision | Large sample size ($N$ = 779), however, 66% of the included exercise interventions had 95% CI overlap 0. DOWNGRADE | Large sample size ($N$ = 778), however, 75% of the included exercise interventions had 95% CI overlap 0. DOWNGRADE | Large sample size ($N$ = 647), however, 65% of the included exercise interventions had 95% CI overlap 0. DOWNGRADE |
| Small study effects | An exhaustive approach was used during the search strategy (i.e., several scientific databases). Egger's test: Z = −1.20 ($p$ = 0.23). Visual inspection of the funnel plots, data extraction sheets and Tables C and D of S5 File revealed no noticeable small study effects. NO DOWNGRADE | An exhaustive approach was used during the search strategy (i.e., several scientific databases). Egger's test: Z = −0.44 ($p$ = 0.66). Visual inspection of the funnel plots, data extraction sheets and Tables C and D of S5 File revealed no noticeable small study effects. NO DOWNGRADE | An exhaustive approach was used during the search strategy (i.e., several scientific databases). Egger's test: Z = 0.76 ($p$ = 0.45). Visual inspection of the funnel plots, data extraction sheets and Tables C and D of S5 File revealed no noticeable small study effects. NO DOWNGRADE |
| Higher quality criteria | | | |
| Large effect | Yes Z = 9.90 ($p$ < 0.001) NO UPGRADE | Yes Z = 9.80 ($p$ < 0.001) NO UPGRADE | Yes Z = 9.07 ($p$ < 0.001) NO UPGRADE |
| Dose response | No clear dose response. NO UPGRADE | No clear dose response. NO UPGRADE | No clear dose response. NO UPGRADE |

(*Continued*)

**Table 2.** (Continued)

|  | A$\dot{V}O_{2peak}$ (L/min) | R$\dot{V}O_{2peak}$ (mL/kg/min) | PPO (W) |
|---|---|---|---|
| Residual confounding | Mixture of exercise modalities, levels of injury, etc. NO UPGRADE | Mixture of exercise modalities, levels of injury, etc. NO UPGRADE | Mixture of exercise modalities, levels of injury, etc. NO UPGRADE |

GRADE certainty in the evidence can be "High," "Moderate," "Low" or "Very Low" according to published guidelines [34]. Judgement on risk of bias followed Cochrane guidance whereby certainty would be downgraded if the reviewers judged that the high risk of bias studies had influenced the pooled effects. Wide variance of point estimates, minimal or no overlap of confidence intervals, and heterogeneity were included as criteria of inconsistency. For heterogeneity, an outcome with $I^2$ >75% was classed as considerable and resulted in a downgrade. Imprecision was downgraded where >50% of studies had confidence intervals overlap the no effect line. Indirectness would have been downgraded where <50% of studies did not include the appropriate main outcome measure or assess a range of participant characteristics. Overall effect sizes are presented as Z-scores. Statistical significance accepted as $p < 0.05$.

AIS, American Spinal Injury Association Impairment Scale; A$\dot{V}O_{2peak}$, absolute peak oxygen consumption; CI, confidence intervals; CRF, cardiorespiratory fitness; PPO, peak power output; RCTs, randomised controlled trials; RoB, risk of bias; R$\dot{V}O_{2peak}$, relative peak oxygen consumption; $\dot{V}O_2$, peak oxygen consumption.

individuals with tetraplegia ($N$ = 3 interventions). For comparison, there is a considerably larger evidence-base for studies including only individuals with paraplegia ($N$ = 31 interventions). A recent systematic review suggested that aerobic exercise results in minimal returns on investment in individuals with tetraplegia, with $\dot{V}O_{2peak}$ improving on average only 9% following 10 to 37 weeks of training [106]. However, their review excluded studies with a sample size <10. Consequently, the Dicarlo study [93], which reported a 94% increase in R$\dot{V}O_{2peak}$, was excluded from their analysis. While the inclusion of this study in the current analysis may have augmented the overall effect, as identified by our leave-one-out analysis, there was nothing untoward in this study to suggest a reason for this exaggerated response. Therefore, our findings indicate that exercise improves CRF in individuals with tetraplegia and that the magnitude of change is not significantly different to individuals with paraplegia.

However, the pooled meta-analysis highlights that individuals with paraplegia (16W) are likely to accrue greater absolute changes in PPO than those with tetraplegia (9W). Typically, higher neurological levels of injury result in a loss of trunk control, motor impairments in the upper-limbs, and reduced mechanical efficiency, compared to lower levels of injury [107,108]. Therefore, individuals with tetraplegia may not have the physical or motor capacity to adapt as effectively as individuals with paraplegia, and thus could experience a ceiling effect with training. Indeed, a recent study identified lesion level as a significant predictor of PPO in a group of handcyclists with SCI [109]. To account for baseline motor function differences between individuals with tetraplegia and paraplegia, we determined relative percentage change for studies that included upper-body aerobic exercise interventions only. The relative percentage change was similar between neurological level of injury classifications: 46% tetraplegia ($N$ = 1 interventions) versus 53% paraplegia ($N$ = 9 interventions). While only 1 tetraplegia-only intervention was included in this subgroup analysis [110], normalising for baseline values seems to indicate similar relative magnitudes of change in PPO.

Williams and colleagues [111] recently demonstrated that individuals with a lower level of injury (<T6) significantly improved PPO compared to individuals with a higher level of injury (≥T6), suggesting a potential role of disrupted cardiovascular control in mediating changes in PPO [112,113]. While methods for ameliorating the consequences of reduced sympathetic cardiovascular control typically associated with injuries ≥T6 have been investigated (e.g., abdominal binding [114], lower-body positive pressure [115], and midodrine [116]), the evidence for an improved CRF is still mixed. A recent case report has indicated that epidural spinal cord

stimulation (SCS) can safely and effectively restore cardiovascular control and improve CRF [117]. With an explosion in SCS studies over the last few years [118], particularly including transcutaneous SCS, the pairing of exercise with novel and noninvasive neuromodulatory approaches will likely continue to receive considerable research attention. Future, adequately powered research may want to consider separating participants into paraplegic and tetraplegic groups or dichotomize by injuries above and below T6 to account for differences in sympathetic cardiovascular control. Currently, there is a paucity of studies analysing data in this fashion, which limits our understanding of how neurological level of injury and the degree of impaired sympathetic cardiovascular control influences the magnitude of change in CRF following an exercise intervention. Researchers may want to consider conducting a battery of autonomic nervous system stress tests at baseline (e.g., Valsalva manoeuvre, head-up tilt, sympathetic skin responses [119]) to determine the degree of supraspinal sympathetic disruption rather than relying on a neurological level of injury derived from a motor-sensory examination. This is important as recent research has indicated that cardiovascular instability cannot be predicted by motor-sensory level and completeness of SCI [120].

There were no significant subgroup differences in CRF based on injury severity. However, the subgroup analysis suggests that individuals with a motor-incomplete SCI may not yield PPO improvements of the same magnitude as individuals with a motor-complete SCI. This is most likely due to the majority of motor-incomplete studies implementing gait training as its exercise modality, which we reveal is the least effective modality for improving CRF. The gait training interventions that measured PPO ($N$ = 2 interventions) used ACE as the CPET modality, demonstrating no transfer effect from lower-body to upper-body exercise. During data extraction, reviewers noted poor reporting of injury severity in a number of studies. While this may be due to older studies having used alternative severity scales (e.g., International Stoke Mandeville Games Federation or Frankel), researchers should endeavour to perform an International Standards for Neurological Classification of SCI (ISNCSCI) exam during screening, and subsequently report an AIS grade, to enable better comparisons to be made between injury severities in the future.

Van der Scheer and colleagues [35] concluded that there was high certainty in the evidence that exercise interventions ≥2 weeks can improve CRF in young and middle-aged adults. However, they revealed that there was a lack of studies exploring the effects of exercise in older adults with SCI (>65 years). The oldest mean age included in our review is 57.9 years [121], and thus supports their call for more research to be conducted in the older SCI population. Interestingly, our meta-regression identified that exercise interventions with participants with a higher mean age were associated with smaller changes in $\dot{V}O_{2peak}$, suggesting that older adults do not achieve the same CRF benefits as younger or middle-aged adults. This is not surprising given the progressive physical deconditioning that occurs naturally with age, as previously shown in the non-injured population [122]. Research indicates that SCI represents a model of advanced ageing [123], with the ageing process being exacerbated in individuals with SCI possibly due to diminished mobility independence resulting in physical deconditioning. Older adults with SCI find it harder to change body position, transfer, and move around independently in comparison to younger adults with SCI [124]. Moreover, it has been suggested that older adults with SCI do not perform volumes or intensities of leisure time physical activity required to achieve fitness benefits [125]. These changes will likely result in reduced incidental physical activity outside of a prescribed exercise intervention. Ageing skeletal muscle is also susceptible to mitochondrial dysfunction, which may be related to chronic inflammation (e.g., "inflammaging"), possibly explaining the diminished responses in CRF for older adults with SCI. Future research may want to investigate optimal strategies for improving CRF in

older adults. Moreover, there is a need for more longitudinal studies that explore the age-related decline in CRF in the SCI population and whether this is accelerated relative to the non-injured population.

Despite a number of recent reviews summarising the effects of specific exercise modalities on the change in CRF following SCI, including aerobic ACE [126], FES-cycling [36], and aerobic plus muscle strength training (mixed multimodal) interventions [127], to the best of our knowledge our pooled meta-analysis is the first to directly compare changes in CRF across a wide range of exercise modalities in individuals with SCI. This review revealed there was a significant subgroup difference in $R\dot{V}O_{2peak}$, with the greatest changes gained via upper-body aerobic exercise or resistance training. The change in $R\dot{V}O_{2peak}$ for upper-body aerobic exercise in the current review (21%) is equivalent to the average 21% improvement reported in a recent systematic review on the effects of ACE in chronic SCI [126]. While the current review did not exclusively investigate ACE, it is evident that aerobic, volitional upper-body exercise training can improve CRF in individuals with SCI. Activating larger amounts of skeletal muscle mass via FES exercise interventions also appears to improve $R\dot{V}O_{2peak}$, yet it is noteworthy that more accessible and less expensive training modalities such as aerobic and resistance training may yield similar or even greater increases in $R\dot{V}O_{2peak}$, despite utilising less muscle mass. Additionally, $R\dot{V}O_{2peak}$ improves following multimodal/hybrid exercise interventions, which challenges a 2015 review reporting inconclusive findings on the effects of combined upper-body aerobic and muscle strength training on CRF [127]. Yet, as the current review included a wide range of interventions not restricted to the upper-body (e.g., aquatic treadmill [63], hybrid cycling [64,69,128], multimodal exercises [129,130]), it is recommended that more research is conducted to delineate whether the improvements in $R\dot{V}O_{2peak}$ with multimodal/hybrid exercise interventions are due to the combination of upper- and lower-body exercise modalities, or due to concurrent training modalities that predominantly use the upper-body (e.g., aerobic plus muscle strength training). Finally, both gait training and behaviour change interventions appear less effective at improving $R\dot{V}O_{2peak}$ and PPO.

Aerobic, upper-body exercise and resistance training modalities demonstrate the greatest improvements in PPO, by 15W and 20W, respectively. It is perhaps unsurprising that resistance training resulted in the largest change in PPO given that these interventions included upper-body exercises prescribed to increase muscular strength, as shown by Jacobs and colleagues [131]. Volitional exercise, as opposed to activity-based therapy modalities (i.e., interventions that provide activation of the neuromuscular system below the level of lesion with the goal of retraining the nervous system, such as FES and gait training), may therefore be more beneficial at improving PPO. Ultimately, these improvements have important ramifications for individuals with SCI that are dependent on performing functional upper-body movements such as transfers or wheelchair propulsion [99,103] and may lead to increased quality of life with more functional independence [132].

Several studies directly compared the effects of specific exercise modalities on the change in CRF [42,63,86]. Notably, Gorman and colleagues [63] demonstrated that there were no transfer effects from a robotic treadmill exercise intervention to ACE performance in a CPET. This review also demonstrates a trend for greater changes in $R\dot{V}O_{2peak}$ when the CPET modality is matched to the intervention (S8 File). Therefore, researchers should endeavour to match the CPET modality to their chosen exercise intervention or at the very least be careful when interpreting changes in CRF when using different modalities.

The current SCI-specific exercise guidelines recommend that exercise should be performed at a moderate-to-vigorous intensity [21]. A recent overview of systematic reviews also advocated the use of moderate-to-vigorous intensity for improving aerobic fitness [133]. The

current pooled meta-analysis demonstrates robust improvements across all CRF outcomes for interventions prescribing exercise at this particular intensity. Furthermore, the additional analysis including cross-sectional studies reveals significant associations of a greater magnitude between MVPA and CRF, as compared to lower-intensity activity. Despite this, our classification of moderate-to-vigorous exercise intensity spans 2 of the ACSM exercise intensity thresholds (S2 File). There may be considerable variation in the actual intensity performed by participants given the noticeable range across the 2 thresholds (e.g., 46% to 90% $\dot{V}O_{2peak}$, 64% to 95% $HR_{peak}$, 12 to 17 RPE). Therefore, individuals with SCI and exercise practitioners should be cautious when prescribing such a broad exercise intensity.

The additional meta-analysis comparing RCT exercise intensities reveals similar changes in CRF outcomes between moderate- and vigorous-intensity interventions. This is in agreement with a previous review [30] and supports the viewpoint from a special communication on high-intensity interval training (HIIT) [31], which suggested that vigorous-intensity exercise is more time efficient and may result in similar if not superior CRF and skeletal muscle oxidative capacity improvements in comparison to moderate-intensity exercise. Interestingly, in a response to a Letter-to-the-Editor [27], the SCI-specific exercise guideline developers acknowledge the need for shorter, effective protocols to be documented in the literature [134]. Furthermore, recent evidence has suggested that HIIT may be more enjoyable than moderate-intensity exercise for individuals with SCI [135], and so this form of training may offer a more time efficient and readily available alternative to moderate-intensity protocols. In the current review, a number of HIIT-based ACE, wheelchair propulsion, and FES hybrid cycling and skiing interventions demonstrated improvements in CRF [69,92,128,136–140]. The potential of "real-world" strategies such as virtual HIIT have also been discussed in a recent review by McMillan and colleagues [141]. Future exercise interventions should look to compare the effects of different HIIT modalities/protocols (i.e., virtual home-based HIIT versus supervised arm-crank HIIT, the most appropriate number and/or length of intervals and recovery periods, vigorous or supramaximal exercise intensities) before concrete recommendations can be made on the most optimal HIIT prescription for improving CRF. While efforts are ongoing to corroborate the safety and feasibility of HIIT [32,92,142], both researchers and individuals with SCI should be vigilant in identifying risks associated with HIIT such as shoulder discomfort or pain, skin irritation or pressure sores caused by abrasive movements, or increased spasticity, along with monitoring for post-exercise hypotension [143]. As higher-intensity exercise generates a greater metabolic heat load than lower-intensity exercise, which increases core body temperature, considerations should be made for individuals with higher neurological levels of injury exercising at a high-intensity and/or in warm to hot ambient temperatures given their greater likelihood of experiencing thermoregulatory issues [144]. General contraindications for performing HIIT have been discussed in the non-injured population [145] yet they also apply for individuals with SCI.

This review reveals that $R\dot{V}O_{2peak}$ and PPO improve regardless of the method used to prescribe exercise intensity. However, with regards to PPO, although not significant the subgroup analysis indicates that the magnitude of change is greater when prescribing intensity via indices of HR (i.e., %$HR_{peak}$, %$HR_{max}$, %HRR) or $\dot{V}O_2$ (i.e., %$\dot{V}O_{2peak}$, %$\dot{V}O_{2reserve}$), compared to RPE and workload. A recent systematic review concluded that exercise interventions using RPE to prescribe relative exercise intensities improved PPO in individuals with SCI [146]. Previous research has revealed that RPE results in inter-individual responses to exercise, with the potential for 2 individuals to perform the same bout of exercise either above or below lactate threshold despite being prescribed the same perceptually regulated intensity, which prevents the development of SCI-specific RPE recommendations [147]. These differences identified in

our meta-analysis may also be due to individuals with SCI being unaccustomed to subjective measures of exertion. Accordingly, there have been calls for better reporting of the standardisation and familiarisation procedures used for RPE [146] and its use has only tentatively been recommended until the evidence base is expanded [148]. Therefore, it seems plausible to suggest that the blunted improvements in PPO with intensity prescribed via RPE, as compared to other prescription methods, may have resulted from insufficient familiarisation before an exercise intervention.

Although HR and $\dot{V}O_2$ have long been used to prescribe exercise intensity, these approaches can result in large training ranges and ignore individual metabolic responses. Particularly, issues may arise with using HR for individuals with a neurological level of injury $\geq$T6, given that these individuals typically exhibit a lower $HR_{peak}$ [149]. The use of fixed percentages (i.e., $\%HR_{peak}$, $\%\dot{V}O_{2peak}$) in the non-injured population has been questioned [150] and has recently been investigated in individuals with SCI, whereby Hutchinson and colleagues [56] showed that fixed $\%HR_{peak}$ and $\%\dot{V}O_{2peak}$ could not guarantee a homogenous domain-specific exercise intensity prescription. Notably, individuals were spread across moderate, heavy, and severe domains at the "moderate" and "vigorous" intensity classifications; thereby questioning whether the "moderate-to-vigorous" terminology used in the SCI-specific exercise guidelines is suitable for adults with SCI.

Given that prescribing exercise intensity via HR and $\dot{V}O_2$ can typically be resource and cost-intensive, there is some scope for using RPE as a cheaper and more practical method for community-based exercise prescription. However, this may not be as effective as other objective methods. Future research should aim to identify the optimal methods of exercise intensity prescription, as well as consider revisiting the current "moderate-to-vigorous intensity" recommendations. Moreover, further research may want to consider using traditional intensity anchors (e.g., the gas exchange threshold, critical power, or lactate threshold) rather than prescribing exercise relative to physiological thresholds to see whether this results in more homogeneous CRF responses to exercise, given research in non-injured individuals suggests this may increase the precision of exercise intensity prescription [151,152]. However, it is worth acknowledging that it is not always possible to identify such traditional intensity anchors in individuals with higher levels of SCI [153,154].

Subgroup analyses based on frequency of sessions and exercise volume reveal no differences in $\dot{RVO}_{2peak}$, thereby supporting the minimal volume of exercise required to attain CRF benefits in individuals with SCI. Furthermore, although there are no statistically significant subgroup differences for PPO based on exercise volume, there is a greater magnitude of change observed for individuals exercising 90 to 149 min/wk in comparison to 40 to 89 min/wk (12W versus 7W change, respectively). A greater weekly exercise volume may therefore accrue greater changes in PPO and, as already described, may be important in improving the capacity to perform daily tasks such as bed or wheelchair transfers [99,103]. Although changes in $\dot{V}O_{2peak}$ are relatively similar between each exercise volume subgroup, and thus corresponding exercise guidelines, the additional meta-analysis on cross-sectional cohorts indicates a significant cumulative impact of prolonged participation in physical activity and exercise. To support this point, a sensitivity analysis revealed a larger difference in $\dot{RVO}_{2peak}$ between inactive individuals and elite athletes, compared to between inactive and active individuals, suggesting that those who exercise more exhibit a greater CRF. Indeed, 2 cross-sectional association studies [40,98] reported significantly higher CRF in individuals with SCI that were habitually performing greater volumes of physical activity. Looking forward, longitudinal RCTs with multiple intervention arms would be the best way to explore dose-response changes with regards to differing volumes of exercise, as has been done in the non-injured population [155–158].

Subgroup analysis based on length of intervention indicated that exercise interventions of 12 weeks or less yield greater changes in $\dot{RVO}_{2peak}$ than those lasting >12 weeks. This may be explained by compliance and adherence issues during prolonged interventions (i.e., >12 weeks). Indeed, the 2 behaviour change studies included in this subgroup analysis observed minimal changes in $\dot{RVO}_{2peak}$ following 16 [82] and 24 weeks [39]. This perhaps emphasises the benefit of short and intensive exercise interventions ≤12 weeks as well as the need for supervised exercise sessions in prolonged interventions to ensure compliance and meaningful changes in $\dot{RVO}_{2peak}$.

Adverse events were reported for at least 3.7% of the total included participants, with the majority of events related to skin irritation, pressure sores, or ulcers. Qualitatively, there was no particular exercise modality that suggested an increased risk for an adverse event, but higher-intensity exercise appeared to reveal more adverse events, albeit being swayed by 1 study in particular [130]. Reporting was poor in a number of studies with reviewers at times unable to determine the exact number of events per participant. Furthermore, there is generally a lack of follow-up assessments following exercise interventions, so it is currently unknown whether there are any detrimental long-term effects of exercise in the SCI population. Going forward, researchers are encouraged to follow a standardised adverse event reporting procedure (including serious and nonserious adverse events) and ensure that they are transparent with reporting of both the nature and the total numbers of each event, either related or unrelated to the exercise intervention.

## 4.1. Strengths and limitations of the review and future directions

**4.1.1. Limitations of the included studies.** Poor reporting of injury characteristics and exercise parameters prevented a perfect comparison of exercise interventions. Overall, studies could have provided more precise descriptions of training parameters to aid with any future refinements to the SCI-specific exercise guidelines. Reporting of adherence to interventions was also poor and should be encouraged to provide an indication of the feasibility or applicability of specific exercise interventions for individuals with SCI. Moreover, adverse events should be transparently reported, even if none occur so that practitioners are able to identify forms of exercise that are most likely to be safe for this population. Additionally, studies typically failed to utilise the training principle of progression, which during prolonged exercise interventions is essential for preventing a plateau in training adaptations and perhaps particularly important in this population for supporting the transition from an inactive lifestyle to higher levels of activity, and ultimately achieving greater CRF benefits [24]. On the whole, the reporting of $\dot{V}O_{2peak}$ attainment criteria was poor, with only 24% of the included exercise interventions using at least 2 criterion methods for identifying when an individual had reached peak capacity, as recently recommended by Alrashidi and colleagues [159]. At least 2 methods [e.g., RPE ≥17, respiratory exchange ratio (RER) ≥1.1, plateau in oxygen uptake] should be adopted for confirming the attainment of a true $\dot{V}O_{2peak}$ to prevent magnitudes of change in CRF from being inflated or underestimated. Furthermore, to the best of our knowledge, only 30% of interventions had a prospectively registered clinical trial entry and only 7.7% had a protocol manuscript published. To sustain the integrity and transparency of reporting in this field, researchers are encouraged to prospectively register any planned clinical trials using publicly available repositories.

The risk of bias assessments on pre-post studies revealed that no study conducted multiple baseline or follow-up assessments. While often time-consuming and impractical with larger sample sizes, multiple assessments ensure reproducibility by accounting for any technical or biological variation, as shown previously in non-injured individuals at risk for type 2 diabetes

[160]. In the SCI population, individuals are typically deconditioned and often exhibit variable responses to a CPET. This variance may be explained by profound blood pressure instability [161], including unintentional "boosting" via episodes of autonomic dysreflexia [162]. Researchers should therefore consider performing multiple CPETs at baseline and follow-up to attain reliable assessments of CRF.

There are also several limitations with regards to the studies included in the additional meta-analyses for this review. First, there is only 1 cross-sectional study using a research-grade wearable device to investigate the association between physical activity and CRF [98]. While self-report questionnaires are valid tools for estimating levels of physical activity [97,163–165], there are important drawbacks including the difficulty of accurately capturing intensity, lack of questionnaires measuring activities of daily living, and recall bias. Secondly, there is a lack of RCTs comparing near-maximal, maximal, or supramaximal exercise intensities to moderate-intensity exercise. The only supramaximal intervention included in this review demonstrated a 17W improvement in PPO [139]. The inclusion of more RCTs comparing vigorous-intensity to lower-intensity exercise could identify whether there are, in fact, benefits to performing shorter but more vigorous-intensity exercise bouts, in comparison to longer continuous forms of exercise.

**4.1.2. Strengths and limitations of the review.** A major strength of the current study is that we pre-planned and prospectively registered (PROSPERO ID CRD42018104342) our systematic review. We used GRADE to assess the certainty in the body of evidence and used quality appraisal tools for the specific study designs included in this review. Our GRADE assessment demonstrates generalisability within the SCI population, through the inclusion of participants across the lifespan and with a wide range of injury characteristics. Yet, the "Low" confidence in the evidence for $A\dot{V}O_{2peak}$ and PPO emphasises the need for more rigorous RCT exercise interventions to address current gaps in the literature [35]. The disparity in GRADE confidence ratings across the specific CRF outcomes is likely a factor of the variability of the total number of included interventions across outcomes $A\dot{V}O_{2peak}$ ($N = 74$), $R\dot{V}O_{2peak}$ ($N = 79$), and PPO ($N = 65$).

As there were not enough RCTs to perform subgroup comparisons and a meta-regression on this study design specifically, we pooled pre-post ($N = 81$) and RCT ($N = 36$) exercise interventions. This is in accordance with Cochrane guidance stating that the inclusion of non-randomised study designs is justified when there are only a small number of RCTs available to provide evidence on the effects of interventions [51]. To consider this limitation, we generated firepower plots to explore the statistical power of the included RCTs and pre-post studies. These plots demonstrate that the RCTs had greater median statistical power across CRF outcomes and were designed to reliably detect a wider range of effect sizes than the pre-post studies. However, the changes in $R\dot{V}O_{2peak}$ and PPO in the primary meta-analysis of RCT interventions relative to controls (2.9 mL/kg/min and 9W, respectively) are similar to those reported in the pooled meta-analysis (2.8 mL/kg/min and 11W, respectively), and thus confirms the robustness of our overall findings. Furthermore, our rigorous approach of adjusting for multiple comparisons minimises any erroneous interpretations of subgroup differences and therefore strengthens our conclusions on the available evidence.

Despite this, the categorisation of interventions within each subgroup could be considered a limitation of the current review. While this was done to directly compare the effects of different subgroups (i.e., acute versus chronic, tetraplegia versus paraplegia, aerobic versus resistance versus FES), it resulted in an unequal number of interventions within each classification and likely underpowered the subgroup comparisons. For example, the lack of a significant difference in the subgroup analysis based on exercise intensity for PPO may be influenced by the

small number of interventions for light- and supramaximal-intensity. Despite reporting some significant subgroup differences across dichotomised studies, these variables were not identified as significant moderator variables in the random-effects meta-regression, meaning these findings should be viewed with caution. It is perhaps more of a limitation of the evidence-base per se, rather than our meta-analysis, in that more RCTs should be conducted to increase the power of these subgroups. Another limitation is that despite our comprehensive search strategy, we may have missed relevant studies as abstracts, theses, and other unpublished work were not included.

## 4.2. Clinical implications and future directions

Our results support the current guidelines regarding the minimal weekly volume of exercise necessary to improve CRF in the SCI population. However, our pooled analysis indicates subgroup differences for PPO based on certain exercise intervention parameters. To the best of our knowledge, there are no large-scale epidemiological studies investigating the dose-response relationship between physical activity and CRF in this population using sensitive and validated methods to quantify the exposure variable (e.g., free-living physical activity). Such studies have been performed in non-injured individuals [166,167]. To identify the optimal stimulus for beneficial CRF responses in this population, dose-ranging studies, akin to those that are used in the pharmaceutical industry, should be conducted. A recent overview of systematic reviews [168] highlighted the poor reporting in exercise interventions in health and disease and called upon the inclusion of checklists [e.g., the Consensus on Exercise Reporting Template (CERT) [169] or the Template for Intervention Description and Replication (TIDieR) [170]] to improve study quality. This would ultimately lead to a better understanding of the "dose" of exercise as medicine required to optimise CRF outcomes in this population.

Both the primary meta-analysis of RCTs (Δ2.9 mL/kg/min) and the pooled meta-analysis (Δ2.8 mL/kg/min) reveal that exercise interventions >2 weeks result in an overall increase in $R\dot{V}O_{2peak}$, which is roughly equivalent to 1 MET-SCI [metabolic equivalent in SCI (2.7 mL/kg/min)] [55]. An increase in maximal aerobic capacity (an estimate of CRF) by 1 MET (3.5 mL/kg/min) in non-injured individuals is associated with a 13% and 15% reduction in all-cause and cardiovascular-related mortality, respectively [171]. The current review shows that individuals meeting the SCI-specific guidelines for cardiometabolic health [21] can improve $R\dot{V}O_{2peak}$ to a similar magnitude to the overall pooled effect (approximately 1 MET-SCI), highlighting that these guidelines may offer a reduction in CVD risk, and therefore mortality. Nonetheless, an association between an improvement in CRF and a reduction in mortality is yet to be established specifically in the SCI population and remains an important avenue of research for the future.

## 5. Conclusion

This systematic review with meta-analysis and meta-regression provides an updated, evidence-based summary of exercise interventions lasting >2 weeks and changes in CRF in individuals with SCI. Based on evidence of low-to-moderate certainty, exercise interventions >2 weeks are associated with significant improvements in CRF, and in particular, a clinically meaningful change in $R\dot{V}O_{2peak}$. Subgroup comparisons from the pooled meta-analysis identified that upper-body, aerobic exercise and resistance training appear the most effective at improving $R\dot{V}O_{2peak}$ and PPO. Furthermore, acutely injured, paraplegic individuals, exercising for more than 3 sessions/week will likely experience the greatest change in PPO. Exercise interventions up to 12 weeks are also most likely to lead to improvements in $R\dot{V}O_{2peak}$. The

meta-regression revealed that older adults may experience smaller changes in $\dot{V}O_{2peak}$ following an exercise intervention. Importantly, there is an ever-growing need for studies to establish a dose-response relationship between exercise and CRF in the SCI population to determine the most optimal form of exercise prescription to reduce the wide-ranging consequences typically associated with SCI. To improve the certainty of evidence in the field moving forward, we call for the development of an SCI-specific reporting template for exercise interventions, as well as encourage researchers to pre-register and/or publish protocol papers for prospective clinical exercise trials. Researchers should also consider whether injury characteristics or participant demographics (e.g., impact of neurological level of injury/severity, motor-sensory classification, autonomic cardiovascular control, age, and sex) influence changes in CRF outcomes with a period of exercise training.

## Supporting information

**S1 PRIMA Checklist. PRIMA Checklist.**
(DOCX)

**S1 File. Systematic Review Search Strategy.**
(DOCX)

**S2 File. Description of Exercise Intensity Classifications as per the American College of Sports Medicine (ACSM) guidelines.**
(DOCX)

**S3 File. Calculated Correlation Factors.**
(DOCX)

**S4 File. Primary meta-analysis (RCT interventions relative to controls).** (1) Overview of participant demographics and injury characteristics for the pooled RCTs (exercise intervention vs. true-world control or standard of care). (2) Summary of the individual RCTs included in the review. (3) Quality assessment rating for each study using the Cochrane Risk of Bias 2 tool. (4) Forest plots and funnel plots for each CRF outcome.
(DOCX)

**S5 File. Change in CRF outcomes in response to prospective, well-characterised exercise interventions lasting >2 weeks from pre-post and RCT studies (pooled meta-analysis).** (1) Overview of participant demographics and injury characteristics for the pooled exercise interventions from pre-post and RCT studies. (2) Summary statistics of the subgroup analyses. (3) Figure visualising the worldwide coverage of exercise interventions included in the secondary, pooled meta-analysis. (4) Summary of the individual studies included in the review. (5) Forest and funnel plots for change in each CRF outcome for each subgroup comparison (time since injury, neurological level of injury, injury severity, exercise modality, length of intervention, relative exercise intensity, method of exercise intensity prescription, frequency of exercise sessions, and exercise volume). (6) Quality assessment ratings for each pre-post study included in the primary meta-analysis. 7. Risk of bias for each RCT intervention arm included in the primary meta-analysis.
(DOCX)

**S6 File. Adverse events.**
(DOCX)

**S7 File. Sensitivity analyses.**
(DOCX)

**S8 File. CPET vs. exercise intervention modality.** (1) Forest plots for each CRF outcome.
(DOCX)

**S9 File. Gait-training sub-analysis.** (1) Forest plots for each CRF outcome.
(DOCX)

**S10 File. Cross-sectional cohort comparisons summary (secondary meta-analysis).** (1)
Overview of participant demographics and injury characteristics for the pooled cohort comparisons. (2) Summary of the individual studies included in the review. (3) Quality assessment rating for each study using the NIH tool for observational cohort and cross-sectional studies. (4) Forest plots and funnel plots for each CRF outcome. (5) Sensitivity analysis on $R\dot{V}O_{2peak.}$ (6) References.
(DOCX)

**S11 File. Cross-sectional associations between physical activity and CRF outcomes.** (1)
Overview of participant demographics and injury characteristics for the pooled association comparisons. (2) Summary of the individual studies included in the review. (3) Quality assessment rating for each study using the NIH tool for observational cohort and cross-sectional studies. (4) Visualisation of correlation coefficients between physical activity dimensions and CRF outcomes across included studies.
(DOCX)

**S12 File. Observational studies (secondary meta-analysis).** (1) Overview of participant demographics and injury characteristics for the pooled observational studies. (2) Summary of the individual studies included in the review. (3) Quality assessment rating for each study using the NIH tool for observational cohort and cross-sectional studies. (4) Forest plots and funnel plots for each CRF outcome.
(DOCX)

**S13 File. RCTs intensity comparisons (secondary meta-analysis).** (1) Overview of participant demographics and injury characteristics for the pooled RCTs comparing the effects of different exercise intensities. (2) Summary of the individual RCTs comparing exercise intensity included in the review. (3) Quality assessment rating for each study using the Cochrane Risk of Bias 2 tool. (4) Forest plots and funnel plots for each CRF outcome.
(DOCX)

## Acknowledgments

We would like to thank Mr. Sajjad Tavassoly (Faculty of Medicine, UBC, Vancouver, Canada) for his assistance with piloting the initial database search. We would also like to thank Dr. Matthew Querée (Department of Physical Therapy, UBC, Spinal Cord Injury Research Evidence Team, GF Strong Rehabilitation Centre, Vancouver, Canada) for his assistance with the risk of bias assessments. We appreciate the assistance of Mr. Adrian Cheng (ICORD) and Dr. Alex Williams (ICORD), who helped pilot the inclusion/exclusion of articles and supported the use of Photoshop to extract data from figures, respectively. We thank Dr. Sam Weaver (University of Birmingham) for support with the meta-analysis using R. Fig 3 was created using Biorender.com.

## Author Contributions

**Conceptualization:** Matthias Walter, Andrei V. Krassioukov, Tom E. Nightingale.

**Formal analysis:** Daniel D. Hodgkiss, Carole Lunny, Catherine R. Jutzeler.

**Investigation:** Daniel D. Hodgkiss, Gurjeet S. Bhangu, Shin-Yi Chiou, Tom E. Nightingale.

**Project administration:** Daniel D. Hodgkiss, Tom E. Nightingale.

**Software:** Daniel D. Hodgkiss, Carole Lunny, Catherine R. Jutzeler.

**Supervision:** Tom E. Nightingale.

**Visualization:** Daniel D. Hodgkiss, Catherine R. Jutzeler.

**Writing – original draft:** Daniel D. Hodgkiss, Gurjeet S. Bhangu, Carole Lunny, Catherine R. Jutzeler, Shin-Yi Chiou, Matthias Walter, Samuel J. E. Lucas, Andrei V. Krassioukov, Tom E. Nightingale.

**Writing – review & editing:** Daniel D. Hodgkiss, Gurjeet S. Bhangu, Carole Lunny, Catherine R. Jutzeler, Shin-Yi Chiou, Matthias Walter, Samuel J. E. Lucas, Andrei V. Krassioukov, Tom E. Nightingale.

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
