## [Editor Report · Decision Letter 0]

3 Aug 2022

Dear Dr Nightingale, 

Thank you for submitting your manuscript entitled "The impact of physical activity and exercise on aerobic capacity in individuals with spinal cord injury: A systematic review with meta-analysis and meta-regression" for consideration by PLOS Medicine.

Your manuscript has now been evaluated by the PLOS Medicine editorial staff and I am writing to let you know that we would like to send your submission out for external peer review.

Please re-submit your manuscript within two working days, i.e. by Aug 05 2022 11:59PM.

Kind regards,

Beryne Odeny

PLOS Medicine

---

## [Decision Letter · Decision Letter 1]

20 Mar 2023

Dear Dr. Nightingale,

Thank you very much for submitting your manuscript "The impact of physical activity and exercise on aerobic capacity in individuals with spinal cord injury: A systematic review with meta-analysis and meta-regression" (PMEDICINE-D-22-02587R1) for consideration at PLOS Medicine. 

Your paper was evaluated by an associate editor and discussed among all the editors here. It was also discussed with an academic editor with relevant expertise, and sent to independent reviewers, including a statistical reviewer. The reviews are appended at the bottom of this email and any accompanying reviewer attachments can be seen via the link below:

[LINK]

In light of these reviews, I am afraid that we will not be able to accept the manuscript for publication in the journal in its current form, but we would like to consider a revised version that addresses the reviewers' and editors' comments. Obviously we cannot make any decision about publication until we have seen the revised manuscript and your response, and we plan to seek re-review by one or more of the reviewers. 

We hope to receive your revised manuscript by Apr 10 2023 11:59PM. Please email us (plosmedicine@plos.org) if you have any questions or concerns.

We look forward to receiving your revised manuscript. 

Sincerely,

Callam Davidson, PhD

PLOS Medicine

plosmedicine.org

Please revise your title according to PLOS Medicine's style. Your title must be nondeclarative and not a question. It should begin with main concept if possible. "Effect of" should be used only if causality can be inferred, i.e., for an RCT. Please place the study design ("A randomized controlled trial," "A retrospective study," "A modelling study," etc.) in the subtitle (ie, after a colon).

Please report your abstract according to PRISMA for abstracts, following the PLOS Medicine abstract structure (Background, Methods and Findings, Conclusions) 

http://www.plosmedicine.org/article/info:doi/10.1371/journal.pmed.1001419

Please provide the data sources, number of studies included, types of study designs included, and synthesis/appraisal methods.

Abstract Background: Provide the context of why the study is important. The final sentence should clearly state the study question.

Please ensure that all numbers presented in the abstract are present and identical to numbers presented in the main manuscript text.

In the last sentence of the Abstract Methods and Findings section, please describe the main limitation(s) of the study's methodology.

Please delete the ‘Key Points’ and replace them with a short, non-technical Author Summary of your research to make findings accessible to a wide audience that includes both scientists and non-scientists. The Author Summary should immediately follow the Abstract in your revised manuscript. This text is subject to editorial change and should be distinct from the scientific abstract. Please see our author guidelines for more information: https://journals.plos.org/plosmedicine/s/revising-your-manuscript#loc-author-summary

Please provide the completed PRISMA checklist, provided at http://www.equator-network.org/reporting-guidelines/prisma/. 

Please add the following statement, or similar, to the Methods: "This study is reported as per the Preferred Reporting Items for Systematic Reviews and Meta-Analyses (PRISMA) guideline (S1 Checklist)."

Please update your search to the present time.

Please include non-English language sources of studies.

Please present and organize the Discussion as follows: a short, clear summary of the article's findings; what the study adds to existing research and where and why the results may differ from previous research; strengths and limitations of the study; implications and next steps for research, clinical practice, and/or public policy; one-paragraph conclusion.

Please remove the Declarations section (and all subsections) and instead ensure that all the information is captured in the submission form questionnaire in Editorial Manager.

Please use ‘et al’ after listing the first six authors in your references, and remove italic/bold formatting. 

Comments from the reviewers:

Reviewer #1: Dear Authors, 

thank you for preparing this wonderful manuscript. Due to competing interests I am not able to provide any further recommendations. 

However, I would like to point out a few things which I find important and I leave to the Editor whether they will share these comments with you. 

1. Numerous experts are against meta-analysis of pre-post intervention studies (or within group differences from RCTs). The authors appropriately downgraded the Evidence to Low certainty, however, I still believe that these results shouldn't be central in your manuscript.

2. A meta-analysis of RCTs is more appropriate (section 3.2.3. RCTs in your manuscript). You do not discuss high heterogeneity in before-mentioned section (was higher than 80%). I believe that results from RCTs should be presented as primary, with subgroup analysis based on types of interventions/controls used, mean age, or health status of study participants. Meta-regression may be useful as well. Also, I suggest that leave-one-out analysis is implemented. 

3. Conclusions are not reflecting the true nature of summarized evidence. In particular, I suggest to implement statements such as: "based on evidence of low certainty..." when summarizing the most important findings. 

4. Directions for future research should focus on identified literature gaps and methodological issues across studies which were discussed in manuscript

5. I suggest Editors to invite a meta-analysis expert to review the manuscript (or senior epidemiologist). 

Reviewer #2: I confine my remarks to statistical aspects of this paper. These were very well done and I recommend publication.

Reviewer #3: In their manuscript entitled "The impact of physical activity and exercise on aerobic capacity in individuals with spinal cord injury: A systematic review with meta-analysis and meta-regression" the authors made analyses on ninety studies and a total of 1,191 participants to conclude that performing exercise >2 weeks results in significant improvements in absolute V˙O 2peak, relative V˙O 2peak and peak power output (PPO) in individuals with SCI. With caution due to the heterogeneity of the subgroup, the authors also reported variables that could influence the effect of training on the PPO, such as time since injury, level of injury, exercise intensity, or training volume. The meta-analyze is well conduced and the results are correctly and exhaustively reported although figures would have been preffered for the main outcomes. 

I only have the few following comments. 

-why didn't the authors include study with whole-body exercise training such as FES-row training? (for exemple: Wheeler, Arch Phys Med Rehabil 2002 or Qiu, Medicine and Science in Sports and Exercise 2016). It seems to me that those RCTs are missing and the reasosn for not considering this training form is not explain in the methods.

-limitation paragraph: first sentence of Page 28: studies with "more rigorous exercise interventions" already exist (see above). Examples need to be provided and the authors need to explain why they did not include these studies here. 

- Line 837: please recall in the sentence the number of RCTs vs. pre-post studies 

- Third paragraph of limitation: line 847: Multiples comparison analyses increase chances to find a comparison significant. i.e. "subgroup differences for PPO based on TSI, neurological level of injury, exercise modality, exercise intensity, method of exercise intensity prescription, and frequency of sessions". was the p value adjusted for this bias?

Reviewer #4: See attachment

Michael Dewey

Reviewer #5: This study explored the changes in cardiorespiratory fitness of spinal cord injury patients through a meta-analysis of previous literature on different exercise interventions, providing clinical evidence for the formulation of exercise prescriptions for spinal cord injury patients. This is a clinically significant study.

1. The second paragraph of the introduction can be appropriately shortened and the parts that are not related to the research can be deleted. You can consider merging it with the third paragraph.

2. The exclusion criteria in the method must be rewritten, as the exclusion criteria are supplementary explanations to the inclusion criteria, rather than the opposite of the inclusion criteria.

3. Adverse event events are a key focus of attention, and can the results be presented in a chart or graph?

4. The main effect was demonstrated in Tables, why not use a forest plot instead, which will look more intuitive?

5. Table 6 is too large, so some content can be simplified to make the layout look better.

6. This study only mentions CRF as the outcome indicator, why not pay attention to the changes of other functional indicators related to spinal cord injury, such as ASIA grading?

7. How the outcome measures (AV˙ O2peak) or relative V˙ O2peak (RV˙ O2peak) are defined? What is the difference between V˙O2peak and AV˙ O2peak/ RV˙ O2peak？

8. In the conclusion, the author mentioned"prescribed via a percentage of oxygen consumption or heart rate", What are the specific values?

9. The final conclusion should recommend which type of exercise, intensity, and duration should be clarified.

Reviewer #6: See attachment

[LINK]

---

## [Decision Letter · Decision Letter 2]

15 Sep 2023

Dear Dr. Nightingale,

Thank you very much for submitting your manuscript "The effect of exercise on aerobic capacity in individuals with spinal cord injury: A systematic review with meta-analysis and meta-regression" (PMEDICINE-D-22-02587R2) for consideration at PLOS Medicine. 

[LINK]

In light of these reviews, I am afraid that we will not be able to accept the manuscript for publication in the journal in its current form, but we would like to consider a revised version that addresses the reviewers' and editors' comments. At this time we cannot make any decision about publication until we have seen the revised manuscript and your response, and we plan to seek re-review by one or more of the reviewers. 

We expect to receive your revised manuscript by Oct 06 2023 11:59PM. Please email us (plosmedicine@plos.org) if you have any questions or concerns.

We look forward to receiving your revised manuscript. 

Sincerely,

Katrien Janin, 

PLOS Medicine

plosmedicine.org

The editorial team feels strongly that several dimensions remain to be resolved:

RTCs (only), RTCs+ pre-post studies, RTCs + everything - uncontrolled and after study effect size. Please systematically conduct the analysis for each of the subgroup analyses crossed with study type.

Clarification of the target populations 

Clarifications of exposures

Clarify study quality (mixing of grade evidence).

Please conduct meta analysis in a subsequential combination. Regress the effect size estimate across all dimensions. 

Furthermore, please also provide strong rationale why a certain analysis/estimator is potentially elevated and to be viewed as primary analyses.

The editorial team also agrees with the comments of the statistical reviewers, these must be resolved in full before this manuscript can proceed.

Comments from the reviewers:

Reviewer #4: I still disagree with the authors about the desirability of including not reported in the analysis of subgroups. I was not suggesting that they should not be reported but I do not see that analysing them with the other groups is helpful. It is mixing two different concepts, the difference between tetraplegic and paraplegic (for example) and the difference between trialists who report the breakdown and those who do not. Including mixed just muddies the waters further.

Michael Dewey

Reviewer #5: The authors have made a thorough effort to comment and revise regarding to questions raised by the reviewers, and I appreciate the work that has been undertaken. I think it will be clinically beneficial.

[LINK]

---

## [Decision Letter · Decision Letter 3]

26 Oct 2023

Dear Dr. Nightingale,

Thank you very much for re-submitting your manuscript "The effect of exercise on aerobic capacity in individuals with spinal cord injury: A systematic review with meta-analysis and meta-regression" (PMEDICINE-D-22-02587R3) for review by PLOS Medicine.

Once again we apologise for the delay and the untimely responses you received to your queries. I have discussed the paper extensively with my colleagues and I am pleased to say that provided the very minor remaining editorial and production issues are dealt with we are planning to accept the paper for publication in the journal.

The remaining editorial issues that need to be addressed are listed at the end of this email. 

The accompanying reviewer attachments can be seen via the link below.

[LINK]

We expect to receive your revised manuscript within 1 week and hope we can move to an editorial accept at that point. Please email me directly at kjanin@plos.org if you have any questions or concerns.

We ask every co-author listed on the manuscript to fill in a contributing author statement. If any of the co-authors have not filled in the statement, we will remind them to do so when the paper is revised. If all statements are not completed in a timely fashion this could hold up the re-review process. Should there be a problem getting one of your co-authors to fill in a statement we will be in contact. 

If you have any questions in the meantime, please contact me (kjanin@plos.org) or the journal staff on plosmedicine@plos.org.  

We look forward to receiving the revised manuscript by Nov 02 2023 11:59PM.   

Sincerely,

Katrien Janin, PhD

Senior Editor 

PLOS Medicine

plosmedicine.org

Requests from Editors:

Please revise your title. "Effect of" should be used only if causality can be inferred, i.e., for an RCT only. Given your study also includes non RTC studies, please revise your title and only use causal language in relation to the part where you use RTCs only. 

Thank you for including your PRISMA checklist as an SI. Please ensure your use section and paragraph numbers, rather than page numbers (given that there will be no page numbers in the inline publication). 

To help us extend the reach of your research, please provide any X (formerly known as Twitter) handle(s) that would be appropriate to tag, including your own, your co-authors’, your institution, funder, or lab. Please provide us with any handles you wish to be included when we post about this paper. 

Last but not least, please contact me directly at kjanin@plos.org with any questions or queries you may have.

Comments from Reviewers:

Reviewer #4: The authors have addressed my remaining point

Michael Dewey

[LINK]

---

## [Editor Report · Decision Letter 4]

30 Oct 2023

Dear Dr Nightingale, 

On behalf of my colleagues and the Academic Editor, I am pleased to inform you that we have agreed to publish your manuscript "Exercise and aerobic capacity in individuals with spinal cord injury: A systematic review with meta-analysis and meta-regression" (PMEDICINE-D-22-02587R4) in PLOS Medicine.

Sincerely, 

Katrien G. Janin, PhD 

Senior Editor 

PLOS Medicine